# Entity Alignment with Noisy Annotations from Large Language Models

**Shengyuan Chen**
Department of Computing
The Hong Kong Polytechnic University
Hung Hom, Hong Kong SAR
shengyuan.chen@connect.polyu.hk

**Qinggang Zhang**
Department of Computing
The Hong Kong Polytechnic University
Hung Hom, Hong Kong SAR
qinggangg.zhang@connect.polyu.hk

**Junnan Dong**
Department of Computing
The Hong Kong Polytechnic University
Hung Hom, Hong Kong SAR
hanson.dong@connect.polyu.hk

**Wen Hua**
Department of Computing
The Hong Kong Polytechnic University
Hung Hom, Hong Kong SAR
wency.hua@polyu.edu.hk

**Qing Li**
Department of Computing
The Hong Kong Polytechnic University
Hung Hom, Hong Kong SAR
csqli@comp.polyu.edu.hk

**Xiao Huang**
Department of Computing
The Hong Kong Polytechnic University
Hung Hom, Hong Kong SAR
xiaohuang@comp.polyu.edu.hk

## Abstract

Entity alignment (EA) aims to merge two knowledge graphs (KGs) by identifying equivalent entity pairs. While existing methods heavily rely on human-generated labels, it is prohibitively expensive to incorporate cross-domain experts for annotation in real-world scenarios. The advent of Large Language Models (LLMs) presents new avenues for automating EA with annotations, inspired by their comprehensive capability to process semantic information. However, it is nontrivial to directly apply LLMs for EA since the annotation space in real-world KGs is large. LLMs could also generate noisy labels that may mislead the alignment. To this end, we propose a unified framework, LLM4EA, to effectively leverage LLMs for EA. Specifically, we design a novel active learning policy to significantly reduce the annotation space by prioritizing the most valuable entities based on the entire inter-KG and intra-KG structure. Moreover, we introduce an unsupervised label refiner to continuously enhance label accuracy through in-depth probabilistic reasoning. We iteratively optimize the policy based on the feedback from a base EA model. Extensive experiments demonstrate the advantages of LLM4EA on four benchmark datasets in terms of effectiveness, robustness, and efficiency.

## 1 Introduction

Knowledge graphs (KGs) serve as a foundational structure for storing and organizing structured knowledge about entities and their relationships, which facilitates effective and efficient search capabilities across various applications. They have been widely applied in question-answering systems (Dong et al., 2023, 2024c), recommendation systems (Catherine & Cohen, 2016; Chen et al., 2024a), social network analysis (Tang et al., 2008), Natural Language Processing (Weikum & Theobald, 2010), etc. Despite their extensive utility, real-world KGs often suffer from issues

38th Conference on Neural Information Processing Systems (NeurIPS 2024).

such as incompleteness, domain specificity, or language constraints, which limit their effectiveness in cross-disciplinary or multilingual contexts. To address these challenges, entity alignment (EA) aims to merge disparate KGs into a unified, comprehensive knowledge base by identifying and linking equivalent entities across different KGs. For instance, by aligning entities between a financial KG and a legal KG, EA facilitates the understanding of complex relationships, such as identifying the same corporations across the two KGs to assess how legal regulations impact their financial performance. This alignment enables a more nuanced exploration and interrogation of interconnected data, providing richer insights into how entities operate across multiple domains.

Entity alignment models predict the equivalence of two entities by measuring their alignment probability. Specifically, rule-based methods (Suchanek et al., 2012; Jiménez-Ruiz & Cuenca Grau, 2011; Qi et al., 2021) utilize predefined rules or heuristics to update alignment probabilities and propagate alignment labels. Conversely, embedding-based models seek to exploit advanced techniques in graph learning (Li et al., 2024; Liu et al., 2024b,a, 2023), parameterizing these probabilities using similarity scores between entity representations learned through knowledge graph embedding algorithms such as translation models (Chen et al., 2017; Sun et al., 2018) or Graph Convolutional Networks (GCNs) (Wu et al., 2019; Mao et al., 2021; Wang et al., 2018; Huang et al., 2023). However, these methods heavily rely on extensive and accurate seed alignments for training—a requirement that poses significant challenges. The need for substantial, cross-domain knowledge to annotate such alignments often makes their acquisition prohibitively expensive.

Recently, Large Language Models (LLMs) have showcased their superior capability in processing semantic information Dong et al. (2024a), which has significantly advanced various graph learning tasks such as node classification (Chen et al., 2024d), graph reasoning (Zhao et al., 2023a; Chai et al., 2023), recommender systems (Zhou et al., 2022; Wu et al., 2023), SQL query generation (Zhang et al., 2024a), and knowledge graph-based question answering (Wang et al., 2024; Zhang et al., 2024b; Dong et al., 2024b). Their capacity to extract meaningful insights from graph data opens up new possibilities for automating EA. Notably, recent studies (Zhong et al., 2022; Zhao et al., 2023b; Jiang et al., 2024) have explored the use of LLMs in EA, primarily focusing on finetuning a pretrained LLM such as Bert to learn semantic-aware representations, relying on accurate seed alignments as training labels. Yet, the potential of LLMs for label-free EA via in-context learning remains unexplored.

However, directly applying LLMs to automate EA poses significant challenges. Firstly, conventional EA models presume that all annotations are correct; yet, LLMs can generate false labels due to LLMs' inherent randomness and the potential incompleteness or ambiguity in the semantic information of entities. Training an EA model directly on these noisy labels can severely impair the final alignment performance. Secondly, given the vast number of entity pairs, annotation with LLMs would be prohibitively expensive. Maximizing the utility of a limited LLM query budget is essential. Existing solutions such as active learning cannot be directly applied since the annotations are noisy.

In response to the outlined challenges, we introduce LLM4EA, a unified framework designed to effectively learn from noisy pseudo-labels generated by LLMs while dynamically optimizing the utility of a constrained query budget. LLM4EA actively selects source entities based on feedback from a base EA model, focusing on those that significantly reduce uncertainty for both the entities themselves and their neighbors. This approach allocates the query budget to important entities, guided by the intra-KG and inter-KG structure. To manage the noisy pseudo-labels effectively, LLM4EA incorporates an unsupervised label refiner that enhances label accuracy by selecting a subset of confident pseudo-labels through probabilistic reasoning. These refined labels are then utilized to train the base EA model for entity alignment. The confident alignment results inferred by the EA model inform active selection in subsequent iterations, thereby progressively improving the framework's effectiveness in a coherent and integrated manner. Contributions are summarized as follows:

- **Novel LLM-based framework for entity alignment:** We propose LLM4EA, an in-context learning framework that uses an LLM to annotate entity pairs. Leveraging the LLMs' zero-shot learning capability, this framework generates pseudo-labels, providing a foundation for entity alignment without ground truth labels.

- **Unsupervised label refinement:** Our framework introduces an unsupervised label refiner informed by probabilistic reasoning. This component significantly improves the accuracy of LLM-derived pseudo-labels, enabling effective training of entity alignment models.

- **Active sampling module:** We propose an active selection algorithm that dynamically searches entities in the huge annotation space. Guided by feedback from the EA model,

this algorithm adjusts its policy based on inter-KG and intra-KG structures, optimizing the utility of LLM queries and ensuring efficient use of resources.

- **Empirical validation and superior performance:** We rigorously evaluate our framework through extensive experiments and ablation studies. The results demonstrate that LLM4EA not only outperforms baselines by a large margin but also shows robustness and efficiency. Each component of the framework is shown to contribute meaningfully to the overall performance, with clear synergistic effects observed among the components.

## 2 Problem definition

A knowledge graph $\mathcal{G}$ comprises a set of entities $\mathcal{E}$, a set of relations $\mathcal{R}$, and a set of relation triples $\mathcal{T}$ where each triple $(e_h, r, e_t) \in \mathcal{T}$ represents a directional relationship between its head entity and tail entity. Given two KGs $\mathcal{G} = \{\mathcal{E}, \mathcal{R}, \mathcal{T}\}, \mathcal{G}' = \{\mathcal{E}', \mathcal{R}', \mathcal{T}'\}$ and a fixed query budget $\mathcal{B}$ to a Large Language Model, we aim to train an entity alignment model $\theta$ based on the LLM's annotations to infer the matching score $m_\theta(e, e')$ for all entity pairs $\{(e, e'), e \in \mathcal{E}, e' \in \mathcal{E}'\}$. The evaluation process utilizes a ground truth alignment set $\mathcal{A}$ to assess the prediction accuracy for target entities in both directions, i.e.,$(e, ?)$ and $(?, e')$ for each true pair $(e, e') \in \mathcal{A}$, based on the ranked matching scores $m_\theta$. Evaluation metrics are hit@k (where $k \in \{1, 10\}$) and mean reciprocal rank (MRR).

## 3 Entity alignment with noisy annotations from LLMs

We aim to design a framework to perform entity alignment with LLMs. Our design is motivated by the following insights. Firstly, we have a huge search space (the overall annotation space is $O(|\mathcal{E}||\mathcal{E}'|)$) to identify the core entity pairs to annotate. Secondly, we don't know whether the annotated labels are correct or not, because we have no prior knowledge or heuristic of the label distribution. Finally, we perform annotations iteratively, requiring the model to adjust its search policy based on annotation effectiveness, while we have no verifiable feedback of this annotation accuracy.

Based on these insights, we propose LLM4EA—an iterative framework that consists of four interconnected steps in each cycle, as illustrated in Figure 1. Initially, **an active selection** technique optimizes the use of resources by choosing critical source entities that significantly reduce uncertainty for themselves and their neighbors. Subsequently, **an LLM-based annotator** identifies the counterparts for the selected source entities, generating a set of pseudo-labels. Next, **a label refiner** improves label accuracy by eliminating structurally incompatible labels. This process involves formulating a combinatorial optimization problem and utilizing a probabilistic-reasoning-based greedy search algorithm to efficiently find a local-optimal solution. Finally, these refined labels are used to train **a base EA model** for the entity alignment task. The outcomes of the alignment then serve as feedback to inform subsequent rounds of the active selection policy. Further details are provided below.

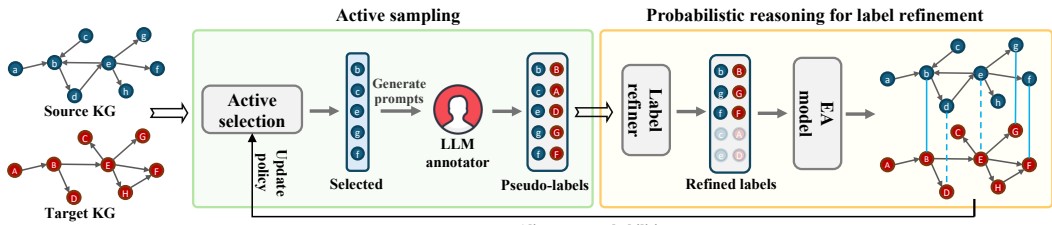

Figure 1: Overview of the LLM4EA framework. LLM4EA utilizes active sampling to select important entities based on feedback from an EA model. It also includes a label refiner to effectively train the base EA model using noisy pseudo-labels. Feedback from the EA model updates the selection policy.

### 3.1 Active selection of source entity

We aim to maximize the utility of the budget by actively allocating the budget to those beneficial entities. To do this, we sample source entities that reduce the most uncertainty of both themselves and their neighboring entities, by a dynamically adjusted policy. The measurement of uncertainty

reduction is based on two assumptions: 1) an entity's own uncertainty is inversely proportional to its alignment probability with its most probable counterpart; 2) the amount of uncertainty an entity eliminates for its neighbors is closely linked to the relational ties between them. To systematically assess this, we introduce the concept of *relational uncertainty*, quantified as follows:

$$U_r(e_h) = (1 - P(e_h)) + \sum_{(e_h, r, e_t) \in \mathcal{T}} w_r \left(1 - P(e_t)\right). \tag{1}$$

Here, $w_r$ is a weight coefficient reflecting the significance of relation $r$ and signifies how much $e_h$ contributes to reducing the uncertainty of $e_t$ through the relation $r$. For this purpose, we employ functionality $\mathcal{F}(r)$ (formally defined in Eq. (4)) as the weight $w_r$, as it quantifies the uniqueness of the tail entity for a given specified head entity. $P(e) := \max_{e' \in \mathcal{E}'} P(e \equiv e')$ represents the alignment probability of the top-match entity for $e$. These alignment probabilities $P(e \equiv e')$ are obtained through probabilistic reasoning during label refinement (Section 3.3.2) and are augmented by the inferred alignments from the base EA model (Section 3.4). In the initial iteration, all alignment probabilities are set to 0.

It's important to note that some source entities are linked to a large number of uncertain neighbors (those with low $P(e)$). These source entities are crucial but may be overlooked if their connected relations have low functionality. Hence, we introduce *neighbor uncertainty* as another metric to assess an entity's importance, by removing the functionality-based weight coefficient:

$$U_n(e_h) = (1 - P(e_h)) + \sum_{(e_h, r, e_t) \in \mathcal{T}} \left(1 - P(e_t)\right). \tag{2}$$

To integrate these two metrics, we employ rank aggregation by mean reciprocal rank:

$$U(e_h) = 2 \times \left( \frac{1}{r_{ur}(e_h)} + \frac{1}{r_{un}(e_h)} \right). \tag{3}$$

Here, $r_{ur}(e_h)$ and $r_{un}(e_h)$ denote the ranking of $e_h$ when using $U_r$ and $U_n$ as metric, respectively. This simple-effective aggregation technique is advantageous for our task since it's scale invariant and requires no validation set for tuning hyperparameters, making it more practical in this task.

### 3.2 LLM as annotator

**Counterpart filtering.** With the selected source entities, we employ an LLM as an annotator to identify the counterpart from $\mathcal{E}'$ for each source entity, generating a set of pseudo-labels $\mathcal{L} = \{(e, e') | e \in \mathcal{E}, e' \in \mathcal{E}'\}$. To narrow down the search space, we first filter out the less likely counterparts before querying the LLM, selecting only the top-$k$ most similar counterparts from $\mathcal{E}'$. The similarity metric is flexible: we use a string matching score based on word edit distance, but other methods are also viable, such as semantic embedding distances derived from word embedding models. By adjusting $k$, we can trade-off between the recall rate of counterparts and the query cost.

**Prompt design.** There are primarily two methods for retrieving context information to construct textual prompts: randomly generated prompts and dynamically tuned prompts. The former involves randomly selecting neighbors to construct contexts for the entity, while the latter dynamically selects neighbors based on feedback from the EA model. For a fair comparison, we use randomly generated prompts across all baselines and the proposed LLM4EA. These prompts include the name of each entity and a set of relation triples to three randomly selected neighbors. For the baseline models, pseudo-labels are generated at once and used for training. For LLM4EA, we evenly divide the budget $\mathcal{B}$ into $n$ iterations and generate pseudo-labels at each iteration using the allocated $\mathcal{B}/n$ budget.

### 3.3 Probabilistic reasoning for label refinement

The pseudo-labels generated by the LLM can be noisy, and directly using these labels to train an entity alignment (EA) model could undermine the final performance. Although estimating the label distribution by asking the LLM for confidence scores or querying multiple times to measure consistency are potential solutions, these approaches can be vulnerable or introduce additional costs.

In light of this, we propose a label refiner that leverages the structure of knowledge graphs. The refinement process is framed as a combinatorial optimization problem aimed at minimizing overall

structural incompatibility among labels. Utilizing a probabilistic reasoning technique, we progressively update our confidence estimation for each label and select those that are mutually compatible, ultimately producing a set of accurate pseudo-labels. Detailed explanations follow below.

### 3.3.1 Functionality and probabilistic reasoning

**Functionality.** The functionality of a relation quantifies the uniqueness of tail entities for a specified head entity, calculated as the ratio of unique head entities to total head-tail pairs linked by the relation. Conversely, inverse functionality quantifies the tail entity uniqueness for a specified head entity. Formally, these are defined as:

$$\mathcal{F}(r) := \frac{|\{e_h|(e_h, r, e_t) \in \mathcal{T}\}}{|\{(e_h, e_t)|(e_h, r, e_t) \in \mathcal{T})\}|}, \quad \mathcal{F}^{-1}(r) := \frac{|\{e_t|(e_h, r, e_t) \in \mathcal{T}\}}{|\{(e_h, e_t)|(e_h, r, e_t) \in \mathcal{T})\}|}. \tag{4}$$

For instance, suppose a KG contains two triples for the relation $locate\_in$: $(Hawaii, locate\_in, US)$ and $(Miami, locate\_in, US)$. Then $\mathcal{F}(locate\_in) = 1.0$ and $\mathcal{F}^{-1}(locate\_in) = 0.5$. In other words, given $(Miami, locate\_in, ?)$, the answer for the missing tail entity is unique; while given $(?, locate\_in, US)$, there are multiple answers for the missing head entity. Such relational patterns are useful for identifying an entity based on its connections within the intra-graph structure.

**Probabilistic reasoning.** If two entities are each connected to entities that are aligned across KGs, this increases the likelihood that they should be aligned as well. Based on this heuristic, an entity pair's alignment probability $P(e_h \equiv e_h')$ can be inferred by aggregating its neighbors' alignment probability via relation functionality:

$$1 - \prod_{\substack{(e_h, r, e_t) \in \mathcal{T}, \\ (e_h', r', e_t') \in \mathcal{T}'}} \left(1 - \mathcal{F}^{-1}(r)P(r \subseteq r')P(e_t \equiv e_t')\right) \times \left(1 - \mathcal{F}^{-1}(r')P(r' \subseteq r)P(e_t \equiv e_t')\right). \tag{5}$$

Here, $P(r \subseteq r')$ denotes the probability of $r$ being a subrelation of $r'$, estimated by alignment probabilities of connected entities:

$$\frac{\sum \left(1 - \prod_{(e_h', r', e_t') \in \mathcal{T}'} \left(1 - P(e_h' \equiv e_h)P(e_t' \equiv e_t)\right)\right)}{\sum \left(1 - \prod_{e_h', e_t' \in \mathcal{E}'} \left(1 - P(e_h' \equiv e_h)P(e_t' \equiv e_t)\right)\right)}. \tag{6}$$

These formulations allow for the propagation and updating of alignment probabilities in a manner that is cognizant of relational structures. We employ this technique to design a label refiner below.

### 3.3.2 Label refiner

**Label incompatibility.** We exploit the "incompatibility" of labels for label refinement, based on the assumption that correct labels can infer each other, while a false label could be incompatible with its correctly aligned neighbors. We define the *overall incompatibility* on a label set $\mathcal{L}$ as:

$$\Phi(\mathcal{L}) := \sum_{(e_h, e_h') \in \mathcal{L}} \left(\mathbf{1}_{P(e_h \equiv e_h') < \max_{e \in \mathcal{E}} P(e, e_h')} + \mathbf{1}_{P(e_h \equiv e_h') < \max_{e' \in \mathcal{E}'} P(e_h, e')}\right). \tag{7}$$

Here, $\mathbf{1}_{P(e_h \equiv e_h') < \max_{e \in \mathcal{E}} P(e, e_h')} = 1$ if $e_h$ is not the top-match for $e_h'$, otherwise 0. It's important to note that a detected incompatibility doesn't necessarily indicate the false alignment of $(e_h, e_h')$: it may suggest a misalignment of their neighbors. Given this, the key to label refinement is to jointly optimize the label's overall incompatibility while avoiding accidentally filtering out correct labels.

**Objective.** To enhance label quality, we propose to refine the pseudo-label set $\mathcal{L}$ by finding a subset $\mathcal{L}^* \subset \mathcal{L}$ that minimizes its overall incompatibility: $\mathcal{L}^* = \arg\min_{\mathcal{L}' \subset \mathcal{L}} \Phi(\mathcal{L}')$. Noteworthy that a trivial solution for this optimization problem is only preserving a set of isolated labels, such that $\max_{e \in \mathcal{E}} P(e \equiv e_h') = 0$ and $\max_{e' \in \mathcal{E}'} P(e_h \equiv e') = 0$ for all $(e_h, e_h') \in \mathcal{L}'$. This trivial solution would lead to the exclusion of most accurate labels, an outcome we aim to avoid. Considering this, we introduce an $l1$ penalty term to penalize the removal of labels, leading to our overall objective:

$$\mathcal{L}^* = \arg\min_{\mathcal{L}' \subset \mathcal{L}} \left(\Phi(\mathcal{L}') + \lambda|\mathcal{L} - \mathcal{L}'|\right). \tag{8}$$

Here $\lambda > 0$ is a weight coefficient. Solving the above combinatorial problem is intractable as it requires computing $\Phi(\mathcal{L}')$ for each possible set $\mathcal{L}' \subset \mathcal{L}$, which is NP-hard. Below we propose to search for a local-optimal solution by a greedy algorithm powered by probabilistic reasoning.

**Greedy search.** The algorithm begins by initializing the alignment probability $P(e \equiv e') = \delta_0$ for every pair $(e, e')$ within the set $\mathcal{L}$, where $\delta_0$ is a constant within the range (0,1). It then iteratively performs a search for an optimal label set $\mathcal{L}'$ through a series of voting steps. Each iteration is comprised of two main steps: probabilistic reasoning and label adjustment.

During the probabilistic reasoning step, the alignment probabilities and subrelation probabilities are updated according to Eq. (5) and Eq. (6), respectively. This update process refines our estimates of label confidence based on the latest information. During the label adjustment step, the label set $\mathcal{L}'$ is updated based on these updated probabilities. Labels are appended to $\mathcal{L}'$ if their updated alignment probabilities exceed $\delta_0$, indicative of high confidence in their alignment, supported by their neighbors:

$$\mathcal{L}' \leftarrow \mathcal{L}' \cup \left\{ (e_h, e'_h) \in \mathcal{L} | P(e_h \equiv e'_h) > \delta_0 \right\}. \tag{9}$$

Conversely, labels demonstrating structural incompatibilities are excluded from $\mathcal{L}'$:

$$\mathcal{L}' \leftarrow \mathcal{L}' \setminus \left\{ (e_h, e'_h) \in \mathcal{L} \mid P(e_h \equiv e'_h) < \max \left( \max_{e \in \mathcal{E}} P(e, e'_h), \max_{e' \in \mathcal{E}'} P(e_h, e') \right) \right\}. \tag{10}$$

In this manner, labels are removed if they are incompatible with updated aligned neighbors, ensuring the preservation of only the most confident pairs within $\mathcal{L}'$. To further refine the search process in subsequent iterations, we augment all entity alignment probabilities within $\mathcal{L}'$ to a superior score:

$$P(e \equiv e') \leftarrow \max \left( P(e \equiv e'), \delta_1 \right) \quad \text{for each } (e, e') \in \mathcal{L}'. \tag{11}$$

Here $\delta_1 \in (\delta_0, 1)$ serves as a new threshold, elevating the alignment probabilities of confident pairs to foster a more directed and effective search. After $n_{lr}$ iterations, we get a set of confidently selected labels $\mathcal{L}^*$ that have high compatibility. The detailed algorithm is presented in Appendix A.2, and analyses of parameter efficiency and computational efficiency are provided in Appendix A.3.

## 3.4 Entity alignment

With the refined labels, we train an embedding-based EA model to learn structure-aware representations for each entity. After training, the EA model computes a matching score $m_\theta(e, e')$ for each entity pair $(e, e')$ for evaluation. The selection of the base EA model is flexible, tailored to the task requirements. We chose a recently proposed GCN-based model, Dual-AMN (Mao et al., 2021), for its effectiveness and efficiency.

Feedback from the base EA model is crucial for dynamic update of the active selection policy. To generate effective feedback, we infer high-confidence pairs $(e, e')$ with the trained EA model, by selecting the pairs that both entities rank top for each other. These pairs are injected into the probabilistic reasoning system. Similar to the label refinement process, this system initializes with an alignment probability of $\delta_0$ for these pairs and updates the estimation of alignment and subrelation probabilities using Eq. (5) and Eq. (6). The updated probabilities are used to construct the uncertainty terms (i.e., $U_r$ and $U_n$) to inform the active selection policy in subsequent iterations, thereby optimizing the budget utility and improving final performance continuously.

## 4 Experiments

In this section, we conduct experiments to evaluate the effectiveness of our framework. We begin by introducing the experimental settings. Then, we present experiments to answer the following research questions: **RQ1.** How effective is the overall framework? **RQ2.** What is the impact of the choice of LLM on the cost and performance of LLM4EA? **RQ3.** What is the effect of the label refiner? **RQ4.** What is the impact of active selection?

### 4.1 Experimental setting

**Datasets and LLM.** In this study, we use the widely-adopted OpenEA dataset (Sun et al., 2020), including two monolingual datasets (D-W-15K and D-Y-15K) and two cross-lingual datasets (EN-DE-15K and EN-FR-15K). OpenEA comes in two versions: "V1" the normal version, and "V2"

the dense version. We employ "V2" in the experiments in the main text. The LLM version in this experiment is GPT-3.5 (gpt-3.5-turbo-0125) and GPT-4 (gpt-4-turbo-2024-04-09). By default, the overall query budget is $\mathcal{B} = 0.1|\mathcal{E}|$.

**Baselines.** Baseline models include three GCN-based models — GCNAlign (Wang et al., 2018), RDGCN (Wu et al., 2019), Dual-AMN (Mao et al., 2021), and three translation-based models — IMUSE (He et al., 2019), AlignE, BootEA (Sun et al., 2018), Here, BootEA is a variant of AlignE that adopts a bootstrapping strategy, equipped with a label calibration component for improving the accuracy of bootstrapped labels. Baseline models are directly trained on the pseudo-labels generated by the LLM annotator, without label refinement or active selection. Every experiment is repeated three times to report statistics.

**Setup of LLM4EA.** We employ GPT-3.5 as the default LLM due to its cost efficiency. Other parameters are $n = 3$, $n_{lr}$, $k = 20$, $\delta_0 = 0.5$, $\delta_1 = 0.9$.

## 4.2 Results

### 4.2.1 Comprehensive evaluation of entity alignment performance

Table 1: Evaluation of entity alignment performance, measured by Hit@K for $K \in \{1, 10\}$, and Mean Reciprocal Rank (MRR), presented in %. Experiment statistics are computed over three trials.

| | EN-FR-15K | | | EN-DE-15K | | | D-W-15K | | | D-Y-15K | | |
|---|---|---|---|---|---|---|---|---|---|---|---|---|
| | Hit@1 | Hit@10 | MRR | Hit@1 | Hit@10 | MRR | Hit@1 | Hit@10 | MRR | Hit@1 | Hit@10 | MRR |
| **Group1. Entity Alignment with GPT-3.5.** | | | | | | | | | | | | |
| IMUSE | 50.0±0.1 | 72.6±0.8 | 57.5±0.4 | 51.6±4.7 | 75.9±3.9 | 60.5±4.5 | 6.0±0.2 | 14.6±2.5 | 9.0±1.0 | 54.4±2.5 | 78.9±1.1 | 63.2±2.0 |
| AlignE | 6.6±0.3 | 24.5±0.5 | 12.6±0.5 | 6.2±0.3 | 18.4±1.0 | 10.4±0.5 | 8.0±0.9 | 24.0±2.7 | 13.3±1.4 | 50.1±2.0 | 76.6±1.4 | 59.2±1.8 |
| BootEA | 44.8±1.1 | 71.9±1.2 | 54.2±1.2 | 68.1±0.2 | 85.4±0.3 | 74.3±0.2 | 60.8±0.2 | 79.3±0.1 | 67.4±0.2 | 87.8±0.1 | 96.7±0.1 | 91.2±0.1 |
| GCNAlign | 17.4±0.3 | 43.2±0.4 | 25.9±0.3 | 22.2±0.2 | 46.2±1.1 | 30.3±0.3 | 16.9±0.1 | 39.3±0.3 | 24.3±0.1 | 45.3±0.4 | 68.3±0.6 | 53.3±0.5 |
| RDGCN | 69.3±0.3 | 82.5±0.3 | 74.3±0.3 | 73.3±4.3 | 84.6±2.6 | 77.4±3.7 | 79.2±0.7 | 89.7±0.5 | 83.2±0.6 | 82.6±3.7 | 91.9±1.3 | 86.1±2.7 |
| Dual-AMN | 51.9±0.3 | 79.6±0.9 | 61.6±0.5 | 70.5±0.7 | 91.1±0.3 | 78.9±0.6 | 62.0±0.1 | 86.8±0.1 | 71.9±0.1 | 85.8±0.3 | 98.4±0.0 | 91.4±0.1 |
| LLM4EA | **74.2±0.3** | **92.9±0.4** | **81.0±0.3** | **89.1±0.5** | **97.8±0.1** | **92.6±0.3** | **87.5±0.3** | **96.7±0.1** | **90.9±0.2** | **97.7±0.0** | **99.5±0.0** | **98.3±0.0** |
| **Group2. Entity Alignment with GPT-4.** | | | | | | | | | | | | |
| IMUSE | 52.7±0.9 | 74.9±1.0 | 59.8±0.9 | 59.6±2.6 | 81.8±1.5 | 67.9±2.1 | 21.6±6.1 | 50.0±10.0 | 31.1±7.4 | 86.6±0.5 | 94.2±0.1 | 89.2±0.4 |
| AlignE | 30.8±2.4 | 69.1±2.5 | 43.1±2.5 | 46.4±5.2 | 76.5±3.8 | 56.6±4.8 | 36.1±3.7 | 67.8±3.6 | 46.7±3.7 | 86.4±0.9 | 97.0±0.3 | 90.2±0.6 |
| BootEA | 58.2±0.3 | 83.7±0.3 | 67.0±0.3 | 80.5±0.4 | 92.6±0.2 | 84.8±0.3 | 71.6±0.2 | 88.3±0.2 | 77.6±0.2 | 95.0±0.1 | 98.6±0.0 | 96.3±0.1 |
| GCNAlign | 30.6±0.0 | 65.3±0.3 | 42.1±0.2 | 41.9±0.4 | 68.6±0.5 | 51.2±0.4 | 31.3±0.3 | 61.6±0.1 | 41.4±0.2 | 82.6±0.2 | 94.9±0.2 | 87.2±0.1 |
| RDGCN | 72.1±0.2 | 84.5±0.1 | 76.7±0.2 | 74.1±1.1 | 85.1±0.7 | 78.0±1.0 | 82.5±1.1 | 91.4±0.7 | 85.9±1.0 | 85.4±0.9 | 93.2±0.4 | 88.3±0.8 |
| Dual-AMN | 76.7±0.1 | 94.9±0.3 | 83.6±0.2 | 90.7±0.1 | 97.9±0.2 | 93.6±0.1 | 81.5±0.1 | 94.9±0.2 | 86.7±0.1 | 97.5±0.0 | 99.3±0.1 | 98.1±0.0 |
| LLM4EA | **80.2±0.3** | **96.0±0.2** | **86.0±0.2** | **93.1±0.5** | **98.7±0.2** | **95.3±0.3** | **89.8±0.3** | **97.9±0.2** | **92.9±0.3** | **97.9±0.1** | **99.6±0.0** | **98.5±0.1** |

To answer **RQ1** and **RQ2**, we conducted two groups of experiments on OpenEA datasets, using GPT-3.5 and GPT-4 as the annotator, respectively. Results are presented in Table 1. We also investigated the performance-cost comparison between the GPT-3.5 annotator and the GPT-4 annotator, illustrated in Figure 2. To control the randomness introduced by the LLMs, each experiment was repeated three times to report mean and standard deviation. These results lead to several key observations:

**First, LLM4EA surpasses all baseline EA models, which are directly trained on the pseudo-labels, by a large margin.** This can be attributed to 1) our label refiner's capability in filtering out false labels, reducing noise during training and enabling more accurate optimization towards the ground true objective; 2) our active selection component's ability to smartly identify important entities to annotate, which takes full advantage of the fixed query budget.

**Second, using the GPT-4 results in higher performance than using the GPT-3.5 as the annotator.** This observation conforms to the fact that GPT-4 is a more advanced LLM with higher reasoning capacity and stronger semantic analysis, resulting in more precise annotation results and higher recall, thus providing more labels of high quality. We also observe that translation-based models (e.g., AlignE) are sensitive to noisy labels under GPT-3.5, while state-of-the-art GCN-based models (e.g., RDGCN and Dual-AMN) are more robust. BootEA also demonstrates superior performance and robustness, attributed to its bootstrapping technique and enhanced by its capability in calibrating bootstrapped labels. However, its label calibration is only applied to the bootstrapped labels, so it still suffers from the false training labels. Our proposed LLM4EA, on the other hand, refines the label accuracy before training the EA model, thus ensuring more accurate training.

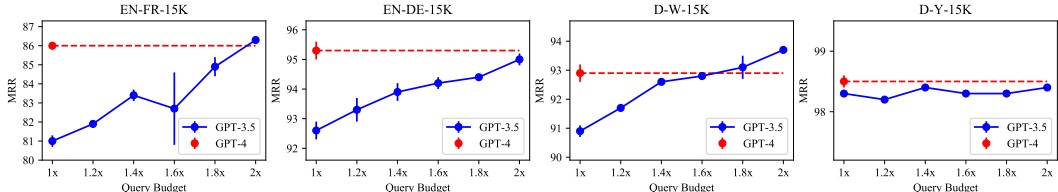

Figure 2: Performance-cost comparison between GPT-3.5 and GPT-4 as the annotator, evaluated by MRR. We increase the budget for GPT-3.5 to evaluate its performance. [$n\times$] denotes using $n\times$ of the default query budget. Each experiment is repeated three times to show mean and standard deviation.

**Finally, LLM4EA is noise adaptive, enabling cost-efficient entity alignment.** To further investigate the effect of the choice of LLM, we examined the performance-cost comparison between GPT-3.5 and GPT-4 as the annotator. We illustrate MRR in Figure 2 (detailed results are available in Appendix B.3). The results show that, by increasing the query budgets (measured by the number of tokens) for GPT-3.5, the performance gradually increases. When the budget is $2\times$ that of GPT-4, the performance is comparable to or exceeds the performance of using GPT-4 as the annotator. According to the pricing scheme of OpenAI, the input/output cost for 1 million tokens for GPT-3.5 and GPT-4 is \$0.50/\$1.5 and \$10/\$30, respectively. This means that our noise-adaptive framework enables cost-efficient entity alignment with less advanced LLMs at $10\times$ less actual cost than using more advanced LLMs, simply by increasing the token budget for the less advanced LLMs.

### 4.2.2 Effect of the label refiner

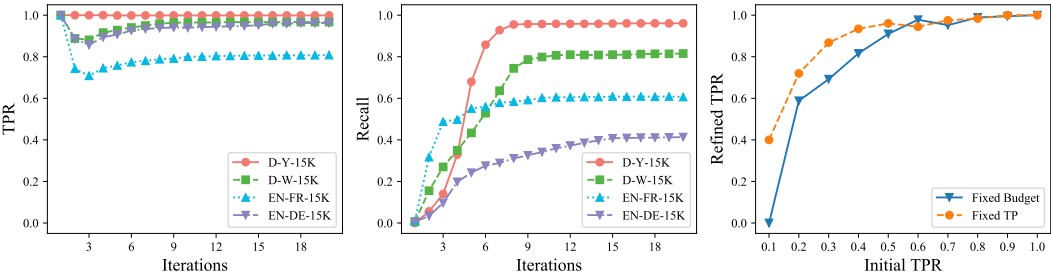

Figure 3: Analysis of the Label Refinement. We illustrate the evolution of the true positive rate (TPR) (left) and recall (middle) for refined labels across four datasets. Furthermore, we assess the robustness of the label refinement process by examining the TPR of refined labels against varying initial TPRs within the D-W-15K dataset (right), with initial pseudo-labels synthesized at different TPR levels.

To answer **RQ3**, we first analyze the evolution of the True Positive Rate (TPR) and the recall rate of the refined labels. Specifically, at each label refinement iteration, the TPR is calculated as $\frac{|\mathcal{A} \cap \mathcal{L}'|}{|\mathcal{L}'|}$, and the recall is calculated as $\frac{|\mathcal{A} \cap \mathcal{L}'|}{|\mathcal{A} \cap \mathcal{L}|}$. The left and middle subfigures of Figure 3 demonstrate how **our label refiner progressively discovers accurate labels and optimizes the TPR.** Initially, the TPR of the refined label set is high (approximately 1.0), then it decreases by a certain percentage, and eventually increases again to a high TPR. We attribute this pattern to: 1) the most confident labels being discovered in the earliest iterations, which are obvious alignments with many connected alignments; 2) as the algorithm progresses, some false pseudo-labels being erroneously added to the label set $\mathcal{L}'$; 3) as the label refinement continues, $\mathcal{L}'$ is adjusted and the false pseudo-labels are replaced with the correct labels inferred by the updated probability as in Eq. (10).

Furthermore, we assess the robustness of our label refiner, as depicted in the right subfigure of Figure 3. We synthesize noisy labels and evaluate the output TPR in relation to varying input TPR levels, using two experimental schemes: *fixed budget*, where the budget remains constant at $0.1|\mathcal{E}|$ while the TPR changes, and *fixed TP*, where the number of true positives is fixed but the TPR and corresponding budgets are adjusted. The results demonstrate that **the label refiner consistently elevates the TPR to over** $0.9$**, even when the initial TPR is around** $0.5$**, showcasing its high**

**robustness to noisy pseudo-labels.** This result also reveals why our framework demonstrates robust performance with the less advanced GPT-3.5 annotator.

### 4.2.3 Ablation study

Table 2: Ablation study overview. The table presents the performance of the LLM4EA (Ours) with various modifications. **Group 1**: removing the label refiner (w/o LR) and the active selection component (w/o Act); **Group 2**: replacing the active selection technique with relational uncertainty (-ru), neighbor uncertainty (-nu), degree (-degree), and functionality sum (-funcSum).

| | EN-FR-15K | | | EN-DE-15K | | | D-W-15K | | | D-Y-15K | | |
|---|---|---|---|---|---|---|---|---|---|---|---|---|
| | Hit@1 | Hit@10 | MRR | Hit@1 | Hit@10 | MRR | Hit@1 | Hit@10 | MRR | Hit@1 | Hit@10 | MRR |
| Ours | 74.2±0.3 | 92.9±0.4 | 81.0±0.3 | **89.1±0.5** | **97.8±0.1** | **92.6±0.3** | 87.5±0.3 | 96.7±0.1 | 90.9±0.2 | 97.7±0.0 | 99.5±0.0 | 98.3±0.0 |
| w/o LR | 51.6±1.0 | 80.2±0.7 | 61.9±0.8 | 74.4±1.7 | 94.2±0.6 | 82.6±1.3 | 39.2±1.4 | 75.7±0.7 | 52.9±1.2 | 85.3±1.0 | 99.2±0.1 | 91.5±0.5 |
| w/o Act | 68.1±2.1 | 88.4±1.7 | 75.4±2.0 | 78.4±0.8 | 93.9±0.3 | 84.6±0.6 | 82.8±0.7 | 92.5±0.6 | 86.3±0.6 | 97.5±0.1 | 99.2±0.3 | 98.1±0.1 |
| Ours-ru | 70.8±1.0 | 91.2±0.4 | 78.2±0.8 | 83.9±0.3 | 97.7±0.2 | 89.6±0.2 | **88.7±0.6** | 97.4±0.3 | **92.1±0.5** | 97.7±0.0 | 99.4±0.1 | 98.3±0.0 |
| Ours-nu | **74.5±0.7** | **93.1±0.5** | **81.2±0.6** | 88.8±0.2 | 96.7±0.3 | 91.8±0.3 | 85.1±0.5 | 95.2±0.5 | 88.9±0.5 | 97.6±0.1 | 99.4±0.0 | 98.2±0.0 |
| Ours-degree | 73.6±2.6 | 92.5±0.8 | 80.4±2.0 | 88.4±0.1 | 96.6±0.2 | 91.5±0.2 | 80.1±3.7 | 90.9±2.2 | 84.0±3.2 | 97.2±0.2 | 99.0±0.1 | 97.9±0.1 |
| Ours-funcSum | 59.5±0.6 | 78.8±0.6 | 66.3±0.6 | 81.2±0.5 | 96.0±0.3 | 87.1±0.4 | 83.9±0.9 | 93.1±1.1 | 87.3±1.0 | 97.5±0.1 | 99.4±0.1 | 98.1±0.1 |

Ablation studies detailed in Table 2 answer **RQ4** and reveal several key insights: **1) Necessity of the Label Refiner for Effective Active Selection:** The performance of "w/o LR", which lacks a label refiner, is inferior not only to other model variants but also to the base model, Dual-AMN. This underscores that active selection depends crucially on reliable feedback, which is compromised when the label refiner is absent; **2) Contribution of Relation and Neighbor Uncertainty in Active Selection:** Incorporating both relation and neighbor uncertainties significantly enhances the utility of the budget. Methods like "Ours-degree" and "Ours-funcSum" focus only on their connections to neighbors and ignore the uncertainty of neighbors. In contrast, "Ours-ru" and "Ours-nu", which take these uncertainties into account, exhibit superior performance. This underscores the importance of considering neighbor uncertainty for effective active selection; **3) Robust Active Selection through Combined Metrics:** Our active selection approach integrates both relation uncertainty and neighbor uncertainty to enable robust active selection. By employing rank aggregation, it prioritizes entities that are deemed significant by both metrics, ensuring a more effective and nuanced selection process.

### 4.2.4 Pareto frontier of runtime overhead against performance.

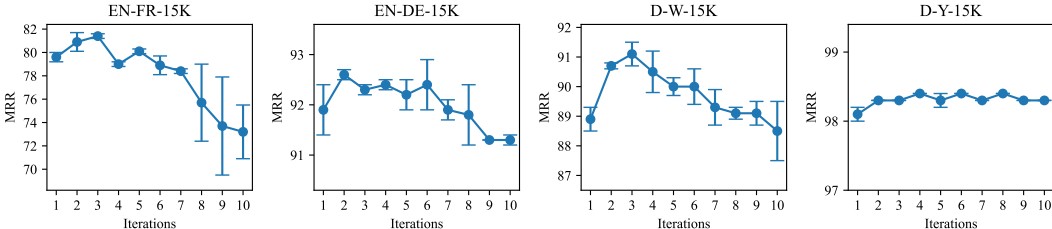

Figure 4: Performance of entity alignment across four datasets with varying active sampling iterations, under a fixed query budget.

The runtime overhead is directly proportional to the number of active selection iterations, $n$, since each iteration involves a subsequent label refinement process. To explore the runtime-performance trade-off in the LLM4EA approach, we examine the Pareto frontier of runtime versus performance. We conduct entity alignment experiments with a fixed query budget, varying the number of active selection iterations. The results of these experiments are illustrated in Figure 4.

The results indicate that performance initially increases as the number of iterations rises from 1 to around 3, but further increases beyond this point lead to a decline. This pattern can be attributed to two main factors: (1) More iterations allow for extensive learning from feedback during the active selection phase. (2) However, when iterations are excessively high, the number of generated pseudo-labels per iteration becomes small, leading to isolated pseudo-labels that undermine the label refinement process. These findings suggest that **an optimal balance between runtime efficiency and performance can be achieved without excessive trade-offs**, indicating a specific threshold for iterations beyond which no further performance gains are observed.

# 5 Related work

**LLMs for Entity Alignment**. Recent approaches have sought to leverage LLMs for entity alignment in knowledge graphs, primarily focusing on integrating structural and semantic information for improved alignment performance. BERT-INT (Tang et al., 2020) fine-tunes a pretrained BERT model to capture interactions and attribute information between entities. Similarly, SDEA (Zhong et al., 2022) employs a pretrained BERT to encode attribute data, while integrating neighbor information via a GRU to enhance structural representation. TEA (Zhao et al., 2023b) reconceptualizes entity alignment as a bidirectional textual entailment task, utilizing pretrained language models to estimate entailment probabilities between unified textual sequences representing entities. A novel approach, ChatEA (Jiang et al., 2024), introduces a KG-code translation module that converts knowledge graph structures into a format comprehensible to LLMs, enabling these models to apply their extensive background knowledge to boost the accuracy of entity alignment. Notably, these models primarily focus on fine-tuning pretrained language models using a set of training labels and do not exploit the zero-shot capabilities of LLMs. In contrast, our proposed model, LLM4EA, leverages the zero-shot potential of LLMs, enabling it to generalize to new datasets without the need for labeled data.

**Probabilistic Reasoning**. In the literature on knowledge graph reasoning, probabilistic reasoning has been effectively applied to infer new soft labels and their associated probabilities from existing labels. It has been utilized in domains such as knowledge graph completion (Qu & Tang, 2019; Zhang et al., 2020; Fang et al., 2023; Chen et al., 2024b) and entity alignment (Suchanek et al., 2012; Qi et al., 2021; Liu et al., 2022; Chen et al., 2024c), where it naturally represents complex relational patterns with simple rules and performs precise inferences. In this work, however, due to the potential inaccuracies in the pseudo-labels generated by LLMs, the newly inferred alignments may be incorrect. Consequently, we opt not to employ probabilistic reasoning directly for the entity alignment task. Instead, we emphasize its use primarily for filtering false pseudo-labels that demonstrate structural incompatibilities within our framework. For completeness, we include the results of comparison with a probabilistic reasoning model – PARIS (Suchanek et al., 2012) in Appendix B.4.

# 6 Limitations

Firstly, during active selection, we distribute the query budget evenly for each selection rather than dynamically customizing the budget allocation for each iteration. This allocation approach could be improved by developing a more adaptive budget allocation strategy. Secondly, the framework currently does not provide direct support for temporal KGs. Although the probabilistic reasoning and active selection components inherently support entity alignment on evolving KGs, the base EA model is transductive. This necessitates retraining the model whenever new entities and relation triples are introduced into the KGs. However, this can be complemented by the research line of inductive entity alignment, such as path-based embedding models or logic-based models, which can generalize to previously unseen entities without the need for retraining.

# 7 Conclusions

In this paper, we address the challenge of automating entity alignment with Large Language Models (LLMs) under budget constraints and noisy annotations. We introduce LLM4EA, a framework that maximizes the utility of a fixed query budget using active sampling and mitigates erroneous labels with a label refiner employing probabilistic reasoning. Empirical results show that LLM4EA's noise-adaptive capabilities reduce costs without sacrificing performance, achieving comparable or superior results with less advanced LLMs at up to 10 times lower cost. This highlights the potential of LLM-based models in merging cross-domain and cross-lingual KGs. Future work will explore incorporating real-time learning capabilities to dynamically adjust to evolving knowledge bases.

## Acknowledgement

The work was fully supported by a grant from the Research Grants Council of the Hong Kong Special Administrative Region, China (Project No. PolyU 15200023).

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

# A Notations and algorithms

## A.1 Notations

Table 3: Notations.

| Notation | Description |
|----------|-------------|
| $\mathcal{G}, \mathcal{G}'$ | The source and target knowledge graphs, respectively |
| $\mathcal{E}, \mathcal{E}'$ | The sets of entities in $\mathcal{G}$ and $\mathcal{G}'$, respectively |
| $\mathcal{R}, \mathcal{R}'$ | The sets of relations in $\mathcal{G}$ and $\mathcal{G}'$, respectively |
| $\mathcal{T}, \mathcal{T}'$ | The sets of relational triplets in $\mathcal{G}$ and $\mathcal{G}'$, respectively |
| $\mathcal{A}$ | The ground truth set of aligned entity pairs |
| $\mathcal{B}$ | The query budget for the LLM |
| $\mathcal{F}(r)$ | The functionality of relation $r$ |
| $P(e \equiv e')$ | The alignment probability of entity pair $(e, e')$ |
| $P(e) = \max_{e' \in \mathcal{E}'} P(e \equiv e')$ | The alignment probability of the top-match entity for $e$ |
| $\mathcal{L}, \mathcal{L}^*$ | The annotated pseudo-label set and the refined pseudo-label set |
| $\Phi(\mathcal{L})$ | The overall incompatibility of the pseudo-label set $\mathcal{L}$ |
| $n$ | The number of iterations of active selection |
| $n_{lr}$ | The number of iterations of label refinement |

## A.2 Pseudo-code of the greedy algorithm

Below we present the pseudo-code of the greedy algorithm, that incorporates probabilistic reasoning to refine the label set.

---
**Algorithm 1** The greedy label refinement algorithm

---
**Inputs:** The pseudo-label set $\mathcal{L}$
**Parameters:** The intialization probability $\delta_0 \in (0, 1)$, the threshold $\delta_1 \in (\delta_0, 1)$, probabilistic reasoning iterations $n_{lr}$
**Outputs:** The refined pseudo-label set $\mathcal{L}^* \subset \mathcal{L}$
$\mathcal{L}' \leftarrow \varnothing$.
$\forall e \in \mathcal{E}, \forall e' \in \mathcal{E}', P(e \equiv e') \leftarrow 0$
$\forall (e, e') \in \mathcal{L}, P(e \equiv e') \leftarrow \delta_0$
$i \leftarrow 0$
**while** $i < n_{lr}$ **do**
$\quad \forall e_h \in \mathcal{E}, \forall e'_h \in \mathcal{E}', P(e_h \equiv e'_h) \leftarrow 1 - \prod_{\substack{(e_h, r, e_t) \in \mathcal{T}, \\ (e'_h, r', e'_t) \in \mathcal{T}'}} \left(1 - \mathcal{F}^{-1}(r) P(r \subseteq r') P(e_t \equiv e'_t)\right) \times$
$\quad \left(1 - \mathcal{F}^{-1}(r') P(r' \subseteq r) P(e_t \equiv e'_t)\right).$     /* Update entity alignment probabilities.*/

$\quad \forall r \in \mathcal{R}, \forall r' \in \mathcal{R}', P(r \subseteq r') \leftarrow \dfrac{\sum\left(1 - \prod_{(e'_h, r', e'_t) \in \mathcal{T}'}\left(1 - P(e'_h \equiv e_h) P(e'_t \equiv e_t)\right)\right)}{\sum\left(1 - \prod_{e'_h, e'_t \in \mathcal{E}'}\left(1 - P(e'_h \equiv e_h) P(e'_t \equiv e_t)\right)\right)}$     /* Update subrelation

$\quad$ probabilities. */
$\quad \mathcal{L}' \leftarrow \mathcal{L}' \cup \{(e_h, e'_h) \in \mathcal{L} | P(e_h \equiv e'_h) > \delta_0\}$ /* Label adjustment, add confident pairs to the label set.*/
$\quad \mathcal{L}' \leftarrow \mathcal{L}' \setminus \{(e_h, e'_h) \in \mathcal{L} \mid P(e_h \equiv e'_h) < \max\left(\max_{e \in \mathcal{E}} P(e, e'_h), \max_{e' \in \mathcal{E}'} P(e_h, e')\right)\}$     /* Label
$\quad$ adjustment, remove less confident pairs from the label set. */
$\quad P(e \equiv e') \leftarrow \max\left(P(e \equiv e'), \delta_1\right) \quad$ for each $(e, e') \in \mathcal{L}'$ /* Elevate alignment probability of confident
$\quad$ pairs. */
**end while**
$\mathcal{L}^* \leftarrow \mathcal{L}' \cup \{(e, e') | P(e \equiv e') > \delta_1\}$     /* Augment the refined label set with confident pairs.*/
**Return** $\mathcal{L}^*$

---

## A.3 Efficient implementation of label refiner

**Parameter-efficient probabilistic reasoning.** The total number of alignment probabilities for all entity pairs is $|\mathcal{E}||\mathcal{E}'|$, resulting in a large parameter size when the KGs involved are extensive. We enhance memory efficiency by adopting a lazy inference strategy in probabilistic reasoning. This

strategy involves only saving the alignment probabilities of the most probable alignments:

$$\left\{ P(e_h, e'_h), |, e_h \in \mathcal{E}, e'_h \in \mathcal{E}', P(e_h \equiv e'_h) = \max\left(\max_{e \in \mathcal{E}} P(e, e'_h), \max_{e' \in \mathcal{E}'} P(e_h, e')\right)\right\}, \quad (12)$$

Probabilities of other entity pairs can be inferred from these saved alignment probabilities using Eq. (5) when necessary. In this way, parameter complexity is reduced to $O(\max(|\mathcal{E}| + |\mathcal{E}'|))$.

**Computation efficiency of probabilistic reasoning.** Probabilistic reasoning is executed iteratively, with each iteration updating the alignment probabilities of all entities and relations following Eq. (5) and Eq. (6). We detail the analysis of these two update phases in the following separately.

Since we adopt a lazy inference strategy, the update of entity alignment probabilities involves updating the set of most probable alignments and associated probabilities as shown in Eq. (12). This update requires the estimation of all $P(e, e'_h), e \in \mathcal{E}$ and all $P(e_h, e'), e' \in \mathcal{E}'$, for each current pair $(e_h, e'_h)$ to determine if this pair needs an update. Consequently, the computational complexity of this process is proportional to the size of this set, which is $O(|\mathcal{E}|)$. The estimated scores $P(e, e'_h), e \in \mathcal{E}$ and $P(e_h, e'), e' \in \mathcal{E}'$ can be precomputed in advance and reused for all pairs $(e_h, e'_h)$, leading to a computation complexity of $O(|\mathcal{E}||\mathcal{E}'|)$. Thus, the overall computational complexity for updating entity alignment probabilities is $O(|\mathcal{E}||\mathcal{E}'| + |\mathcal{E}|) = O(|\mathcal{E}||\mathcal{E}'|)$.

The update process for sub-relation probabilities involves updating all $P(r \subset r')$ for $r \in \mathcal{R}$ and $r' \in \mathcal{R}'$, resulting in a complexity of $O(|\mathcal{R}||\mathcal{R}'|)$. The estimation of Eq. (6) utilizes the probabilities of the most probable alignments from Eq. (12). Notably, most relation pairs $(r, r')$ do not have aligned head entities $(e_h, e'_h)$ or aligned tail entities $(e_t, e'_t)$, thus most relation pairs can be excluded for efficient computation by exploiting this sparsity heuristic, reducing the computations by orders.

It is worth noting that these computations can be further accelerated through parallelization, as their execution solely depends on the results from the previous iteration.

# B  Experimental details

## B.1  Hardware and software configurations

Our experiments were conducted on a server equipped with six NVIDIA GeForce RTX 3090 GPUs, 48 Intel(R) Xeon(R) Silver 4214R CPUs, and 376GB of host memory. The models were implemented using TensorFlow, NumPy, and SciPy. It was observed that the software version significantly affects hardware-software compatibility. Specifically, the original implementation of Dual-AMN was based on TensorFlow 1.14.0, which is not compatible with newer GPUs such as the NVIDIA GeForce RTX 3090. Therefore, we updated the code to be compatible with TensorFlow 2.7.0, enabling the model to leverage GPU acceleration effectively. The details of the software packages used in our experiments are listed in Table 4.

Table 4: Package configurations of our experiments.

| Package | tqdm | numpy | scipy | tensorflow | keras | openai |
|---------|------|-------|-------|------------|-------|--------|
| Version | 4.66.2 | 1.24.4 | 1.10.1 | 2.7.0 | 2.7.0 | 1.30.1 |

## B.2  Dataset statistics and preprocessing

In our experiments, we utilized the OpenEA dataset version 1.1 (V2), specifically the 15K set. The statistics are detailed in Table 5. It's important to highlight that the destination dataset for D-W-15K, originating from Wikidata, contains only entity IDs, lacking explicit names. These IDs, devoid of semantic content, are not inherently meaningful to a language model. To rectify this, we processed the dataset using the 'wikidatawiki-20160801-abstract.xml' dump file from Wikidata. This file provided the raw data necessary for constructing the D-W-15K dataset, enabling us to extract meaningful entity names.

We shown the quantity of entities with names (after name extraction) for each KG in the 'Named entities' column in Table 5.

Table 5: Data statistics of used OpenEA dataset.

| Datasets | KG | Relations | Relation triplets | Attributes | Attribute triples | Named Entities | Targets in top-$k$ |
|---|---|---|---|---|---|---|---|
| EN-FR | English (EN) | 193 | 96,318 | 189 | 66,899 | 15,000 | 13,550 |
| | French (FR) | 166 | 80,112 | 221 | 68,779 | 15,000 | |
| EN-DE | English (EN) | 169 | 84,867 | 171 | 81,988 | 15,000 | 13,330 |
| | German (DE) | 96 | 92,632 | 116 | 186,335 | 15,000 | |
| D-W | DBpedia (DB) | 167 | 73,983 | 175 | 66,813 | 15,000 | 12,910 |
| | Wikidata (WD) | 121 | 83,365 | 457 | 175,686 | 13,458 | |
| D-Y | DBpedia (DB) | 72 | 68,063 | 90 | 65,100 | 15,000 | 15,000 |
| | Yago (YG) | 21 | 60,970 | 20 | 131,151 | 15,000 | |

In the counterpart filtering phase, we selected the top-$k$ (where $k = 20$ for our experiments) most similar candidates. The 'Target in top-$k$' column of Table 5 shows the number of target entities included in this selection.

## B.3 Performance-cost comparison between GPT-3.5 and GPT-4

Table 6: Performance-cost comparison between GPT-3.5 and GPT-4 as annotator. We increase the budget for GPT-3.5 to evaluate its performance. [$n\times$] denotes using $n\times$ of the default query budget.

| | EN-FR-15K | | | EN-DE-15K | | | D-W-15K | | | D-Y-15K | | |
|---|---|---|---|---|---|---|---|---|---|---|---|---|
| | Hit@1 | Hit@10 | MRR | Hit@1 | Hit@10 | MRR | Hit@1 | Hit@10 | MRR | Hit@1 | Hit@10 | MRR |
| GPT-3.5 | 74.2±0.3 | 92.9±0.4 | 81.0±0.3 | 89.1±0.5 | 97.8±0.1 | 92.6±0.3 | 87.5±0.3 | 96.7±0.1 | 90.9±0.2 | 97.7±0.0 | 99.5±0.0 | 98.3±0.0 |
| GPT-3.5 (1.2×) | 75.4±0.4 | 93.2±0.4 | 81.9±0.2 | 90.2±0.6 | 98.0±0.0 | 93.3±0.4 | 88.4±0.1 | 97.1±0.2 | 91.7±0.1 | 97.6±0.0 | 99.3±0.1 | 98.2±0.0 |
| GPT-3.5 (1.4×) | 77.2±0.2 | 94.5±0.5 | 83.4±0.3 | 91.0±0.4 | 98.1±0.1 | 93.9±0.3 | 89.2±0.2 | 97.9±0.0 | 92.6±0.1 | 97.7±0.0 | 99.5±0.0 | 98.4±0.0 |
| GPT-3.5 (1.6×) | 76.3±2.4 | 94.1±0.7 | 82.7±1.9 | 91.4±0.2 | 98.4±0.0 | 94.2±0.2 | 89.6±0.0 | 97.9±0.2 | 92.8±0.1 | 97.7±0.0 | 99.4±0.0 | 98.3±0.0 |
| GPT-3.5 (1.8×) | 78.8±0.5 | 95.2±0.4 | 84.9±0.5 | 91.9±0.1 | 98.1±0.0 | 94.4±0.1 | 90.1±0.5 | 97.9±0.2 | 93.1±0.4 | 97.7±0.0 | 99.5±0.1 | 98.3±0.0 |
| GPT-3.5 (2×) | 80.6±0.2 | 95.9±0.1 | 86.3±0.1 | 92.7±0.2 | 98.5±0.1 | 95.0±0.2 | 90.7±0.2 | 98.5±0.1 | 93.7±0.1 | 97.8±0.0 | 99.5±0.0 | 98.4±0.0 |
| GPT-4 | 80.2±0.3 | 96.0±0.2 | 86.0±0.2 | 93.1±0.5 | 98.7±0.2 | 95.3±0.3 | 89.8±0.3 | 97.9±0.2 | 92.9±0.3 | 97.9±0.1 | 99.6±0.0 | 98.5±0.1 |

## B.4 Performance comparison against rule-based models

Table 7: Performance comparison against rule-based models, evaluated by precision, recall, and f1-score in %.

| | EN-FR-15K | | | EN-DE-15K | | | D-W-15K | | | D-Y-15K | | |
|---|---|---|---|---|---|---|---|---|---|---|---|---|
| | Precision | Recall | F1-score | Precision | Recall | F1-score | Precision | Recall | F1-score | Precision | Recall | F1-score |
| Emb-Match | 88.9 | 73.6 | 80.5 | 89.7 | 69.5 | 78.3 | 91.8 | 62.4 | 74.3 | 100 | 100 | 100 |
| Str-Match | 84.8 | 69.8 | 76.6 | 92.3 | 71.4 | 80.5 | 96.2 | 57.9 | 72.3 | 76.9 | 100 | 86.9 |
| PARIS | 58.3±0.5 | 26.5±0.3 | 36.5±0.4 | 90.8±0.3 | 50.7±0.3 | 65.0±0.3 | 92.4±0.4 | 70.2±0.2 | 79.8±0.2 | 99.1±0.1 | 96.7±0.1 | 97.9±0.1 |
| LLM4EA | 68.6±0.3 | 53.1±0.2 | 59.8±0.2 | 90.5±0.3 | 82.4±0.4 | 86.2±0.4 | 90.7±0.4 | 81.6±0.5 | 85.9±0.5 | 98.9±0.0 | 97.6±0.1 | 98.3±0.0 |

In this section, we compare the LLM4EA model with several rule-based models, including two lexical matching-based approaches: Emb-Match and Str-Match. Emb-Match uses cosine similarities between word embeddings to identify aligned pairs, employing a pretrained language model. Str-Match utilizes the Levenshtein Distance to calculate similarity scores. Additionally, the probabilistic reasoning model PARIS performs entity alignment by relying on probabilistic methods.

For the lexical matching-based models, we compute the similarity and evaluate the confident entity pairs, specifically targeting those whose normalized similarity scores exceed 0.8. The pretrained language model used for Emb-Match is bert-base-uncased for its ability to process unseen words as a subword model. To reduce false positives, we implementing a filtering process that only considers 1-1 matching for embedding-based matching. For PARIS, we assess all inferred aligned pairs by setting the threshold to zero. The entity alignment (EA) model of LLM4EA generates a ranked score list, which is not directly comparable with these rule-based models. To facilitate comparison, we use the trained EA model to generate confident pairs, ensuring that each entity is the top-ranked candidate for its counterpart. These pairs are then processed through our label refiner, and the refined pairs are evaluated. Experiments for PARIS and LLM4EA are repeated three times to ensure statistical reliability; for Emb-Match and Str-Match, experiments are performed once as these algorithms are deterministic. The results are presented in Table 7.

The findings highlight several key observations: 1) Lexical matching-based methods show a degree of alignment ability by leveraging name similarity, particularly on the D-Y-15K dataset, where many aligned entities have identical names. 2) These methods, however, encounter scalability challenges due to their reliance on name similarity. As the size of the KG grows, the number of similar entities increases, making it difficult to maintain precision. This is evident from our experiments on EN-FR-100K and EN-DE-100K, where precision dropped to 49% and recall to 65%, a significant decrease compared to the 15K-size datasets, supporting our analysis. 3) PARIS achieves precise results by handling noisy data through probabilistic reasoning, although its recall is lower than LLM4EA's. 4) LLM4EA, with its active selection and label refinement techniques, consistently delivers robust and accurate performance across all datasets.

