# OpenReview forum: "Entity Alignment with Noisy Annotations from Large Language Models"
_NeurIPS.cc/2024/Conference — NeurIPS 2024 poster_

### Official Review · Reviewer_SZjY · 2024-06-17

**Soundness:** 2
**Presentation:** 3
**Contribution:** 1
**Rating:** 4
**Confidence:** 5

**Summary:**

The paper proposes LLM4EA for annotating entity alignment pairs using an LLM. It introduces an active learning policy and an unsupervised label refiner to efficiently collect pseudo-labels. Experiments on OpenEA benchmarks demonstrate the strong performance of LLM4EA.

**Strengths:**

The paper is well-structured and straightforward, making it easy to understand the design and motivation behind LLM4EA.

Applying an LLM for label annotation is both simple and effective.

The results on OpenEA-V2 significantly outperform the baseline methods.

**Weaknesses:**

The algorithm in this paper somewhat echoes that of BootEA, which limits its novelty. While earlier iterative or bootstrapping algorithms generated pseudo-labels using cosine similarity or other metrics, this paper introduces the use of an LLM to select pseudo-labels. Beyond this aspect, the procedure closely resembles existing methods.

The declaration in Line 237 is incorrect; "V1" is the dataset that resembles real-world KGs, and "V2" is the easier version.

Table 1 is not clear. Is LLM4EA the only method evaluated that considers pseudo-labels generated by an LLM? If so, why are the performances of the other methods significantly lower than those reported in the OpenEA paper?

Utilizing LLMs to annotate labels carries a risk of data leakage, as many entity name pairs in the OpenEA datasets are identical. This flaw was also noted by the creators of OpenEA.
If the authors account for the entity name as a feature, they should also benchmark LLM4EA against approaches that incorporate textual information, for example, MultiKE.

**Questions:**

Please see Weaknesses.

---

> ### Author Rebuttal · Authors · 2024-08-07
>
> Thank you for your constructive feedback, which has greatly helped improve the draft. Below, we address your concerns and weaknesses (w1-w4).
>
> # [w1] Technical Novelty (distinctions from bootEA)
>
> We respectfully point out that the LLM4EA is significantly distinct from bootEA, we outline three major differences between them:
>
> 1. **Pseudo-label generation.** Comparing with the bootEA, our work generates pseudo labels by directly processing semantic information, while bootEA generates pseudo labels based **on similarity scores grounded on embeddings trained on pre-aligned labels**. LLM4EA exploits the potential of automating the EA task in a **label-free** fashion.
> 2. **Unsupervised label refinement**. Our label refiner is unsupervised, while BootEA relies on the embeddings **learned on training labels** to compute the label’s confidence. As a result, **directly adopting BootEA to this task cannot mitigate the initial noisy annotated pseudo-labels**. As we have shown in Table 1 and discussed in line 265-268 in the paper draft, BootEA still suffers from false labels if directly trained on the noisy labels from LLMs.
> 3. **Optimized resource allocation**. We introduce an active search policy to improve the utility of the query budget.
>
>
>
> # [w2 ] Results on OpenEA V1
>
> Thank you for the detailed review and helpful feedback, we will correct the false statement of the dataset version in the revision. To validate the performance on real-world KGs, we attach the experimental results on OpenEA V1 dataset as below. Due to character limit, we only attach results of strong baselines.
>
> |                | D_W       | (w/o ent  | name)     |           | D_Y       |           |           | EN_DE     |           |           | EN_FR     |           |
> | --- | -- | - | -- | -- | -- | -- | -- | -- | -- | -- | -- | - |
> |      | hit@1     | hit@10    | mrr       | hit@1     | hit@10    | mrr       | hit@1     | hit@10    | mrr       | hit@1     | hit@10    | mrr       |
> | BootEA         | 0.302     | 0.584     | 0.413     | 0.664     | 0.811     | 0.718     | 0.655     | 0.844     | 0.720     | 0.429     | 0.685     | 0.515     |
> | Dual-Amn       | 0.386     | 0.639     | 0.479     | 0.748     | 0.858     | 0.790     | 0.728     | 0.910     | 0.793     | 0.537     | 0.800     | 0.628     |
> | RDGCN          | 0.337     | 0.436     | 0.372     | 0.817     | 0.923     | 0.858     | 0.646     | 0.775     | 0.692     | 0.597     | 0.742     | 0.649     |
> | LLM4EA         | **0.472** | **0.654** | **0.536** | 0.755     | 0.859     | 0.795     | **0.736** | **0.918** | **0.800** | 0.581     | **0.819** | 0.662     |
> | LLM4EA (RDGCN) | 0.421     | 0.532     | 0.476     | **0.842** | **0.931** | **0.883** | 0.672     | 0.787     | 0.736     | **0.642** | 0.761     | **0.683** |
>
>
>
>
>
> As the results show, our model consistently outperforms baselines. Noteworthy that LLM4EA is a general framework and can employ any base EA model such as RDGCN (last row) to perform label-free entity alignment. **We kindly argue that our conclusions and claims hold on this more realistic dataset**. And we will include the updated results in the draft revision.
>
> # [w3] Clarification of baseline setting
>
> Allow us to clarify the confusion about the experimental setting in weakness3.
>
> 1.   **The input of baselines are pseudo labels**.
>
> As stated in the experimental setting (line244-245 in the draft), baseline models are also trained on the pseudo-labels generated by the LLMs. This setting ensures **fair comparison** because the inputs are the same (same query budget to the LLMs).
>
>
>
> 2.   **Baselines perform lower than OpenEA's paper for two reasons:**
>
> **1) the annotated pseudo-labels are noisy** and existing methods cannot handle without a pre-refinement of noisy labels before training; and **2) the training label size is smaller** ($0.1|\mathcal{E}|$, but in OpenEA's paper it's $0.2|\mathcal{E}|$ ).
>
> # [w4] name bias concern
>
> The use of name information is often referred to as name bias. We answer this concern from two aspects:
>
> 1.   **When name information is used**: LLM4EA can generate pseudo-labels and enable robust learning. While most existing methods like MultiKE and RDGCN exploit entity names as features, they <u>rely on pre-aligned pairs</u> for training. LLMs can serve as pseudo-label generators.
> 2.   **When names are not available**: LLM4EA still works. By processing semantics within attributes, LLM4EA can generate effective labels. We empirically evaluate this on D_W_15K_V1 (**1st column** in the following table), we masked target entity names with IDs to avoid name bias.  And compare with MultiKE and RDGCN which both leverage semantic features.
>
> |    | D_W       | (w/o ent  | name)  |    | D_Y   |   |    | EN_DE  |   |   | EN_FR   |    |
> | - | - | - | - | - | - | - | - | -- | - | - | -- | -- |
> |      | hit@1     | hit@10    | MRR     | hit@1     | hit@10    | MRR       | hit@1     | hit@10    | MRR | hit@1  | hit@10    | MRR       |
> | MultiKE  | 0.021     | 0.064   | 0.036     | 0.503     | 0.767     | 0.598     | 0.357     | 0.666     | 0.461     | 0.274     | 0.590     | 0.380     |
> | RDGCN   | 0.337     | 0.436   | 0.372     | 0.817     | 0.923     | 0.858     | 0.646     | 0.775     | 0.692     | 0.597     | 0.742     | 0.649     |
> | LLM4EA    | **0.472** | **0.654** | **0.536** | 0.755     | 0.859     | 0.795     | **0.736** | **0.918** | **0.800** | 0.581     | **0.819** | 0.662     |
> | LLM4EA (RDGCN) | 0.421  | 0.532     | 0.476     | **0.842** | **0.931** | **0.883** | 0.672     | 0.787     | 0.736     | **0.642** | 0.761     | **0.683** |
>
> As shown in the above table,
>
> -   LLMs serve as an effective annotator for all methods to perform label-free EA, which justifies the above 1st argument.
> -   LLMs can generate effective labels without name information, which justifies the 2nd claim.
>
> We hope our responses can satisfactorily address your concerns and we would appreciate it if you could consider raising the score.

---

> > ### Comment · Reviewer_SZjY · 2024-08-12
> >
> > Thank you for your detailed response. However, I am still confused and curious about why you chose a smaller training label set and why your new experiments still do not follow the standard setting (i.e., 0.2|E|). The OpenEA datasets have a default setting of 0.2 because it is convenient for 5-fold cross-validation. Were your results based on this setting?
> >
> > The pseudo-label setting for baselines is also unfair. Why do the baselines have to use the pseudo-labels generated by the LLM for training? Methods like BootEA have their own iterative process to refine their pseudo-labels. This is also why I think the proposed method lacks novelty. The authors just replaced their decision function with an LLM.
> >
> > The usage of the name feature is still unclear. How can "LLMs generate effective labels without name information, which justifies the 2nd claim"? In Lines 139-142, the authors claim that the annotation filtering is based on string matching score or word embeddings (this also justifies that your method also generates pseudo-labels based on similarity scores grounded on embeddings). If the string or word does not exist, how do you filter out the less likely counterparts?
> >
> > Then, in Lines 147-148, the authors state that the prompts include the name and relational triplets. If the name feature is unavailable, how can you represent these triplets?

---

> > > ### Author Response · Authors · 2024-08-13
> > > **Response to reviewer SZjY (1)**
> > >
> > > We sincerely thank you for engaging in this discussion and allowing us to address any concerns. Below we first restate our setting and workflow in detail to clarify any confusion about our setting, then answer each of your questions.
> > >
> > > # 1. Settings
> > >
> > > **1.1 All experiments are under the label-free setting without ground truth labels for training**
> > >
> > > Our work is motivated to explore label-free entity alignment, aiming to train EA models using **only LLM-annotated pseudo-labels**. All experiments, including evaluations of both LLM4EA and baseline methods, are conducted **without ground truth labels for training.** This setting is specifically designed to test whether all methods effectively achieve label-free EA. The baselines assess the effectiveness of directly training existing EA models on these noisy pseudo-labels, whereas LLM4EA is a framework designed to mitigate noise and optimize budget allocation for effective learning of the integrated base EA model.
> > >
> > > **1.2 LLM4EA works in 5-steps**
> > >
> > > LLM4EA operates in a five-step process for each iteration:
> > >
> > > Step1. Select important source entities
> > >
> > > Step2. Recall top-k candidate counterparts for each source entity by string match or word embedding matching
> > >
> > > Step3. Use an LLM to identify the target entity from the top-k candidates for each selected source entity.
> > >
> > > Step4. Refine the pseudo-labels by probabilistic reasoning
> > >
> > > Step5. Train the EA model
> > >
> > > We respectfully point out that, although LLM4EA is iterative, this process is not bootstrapping like BootEA. Instead, this iterative manner is introduced to dynamically adjust the source entity selection (step1) policy to maximize the utility of a fixed annotation budget. If the budget is $\mathcal{B}$ and the iteration number is 5, then LLM4EA selects and annotates $\mathcal{B}/5$ source entities at each iteration. **If we remove the active selection module in LLM4EA, the LLM generates the seed alignments at once as training data and has no iterations, as in baselines**.
> > >
> > > # 2. Answers to the questions
> > >
> > > We will first list the questions for clarity and then provide corresponding answers, based on the detailed settings outlined above.
> > >
> > > **2.1 Label ratio and the experimental setting**
> > >
> > > **Q1.** why you chose a smaller training label set and why your new experiments still do not follow the standard setting (i.e., 0.2|E|).?
> > >
> > > **Q2.** Were your results based on this setting(5-fold cross validation)?
> > >
> > > **Q3.** The pseudo-label setting for baselines is also unfair.
> > >
> > > **Q4.** Why do the baselines have to use the pseudo-labels generated by the LLM for training?
> > >
> > > **Answers: **
> > >
> > > - **We did not follow the 5-fold setting of OpenEA because we are under label-free setting.** We undertand the OpenEA dataset by default contains $0.2|\mathcal{E}|$ **ground truth** labels for training,  which is designed for **supervised/semi-supervised setting**. As stated in setting 1.1, our experiments evaluate **label-free** EA with LLMs, the only input to the baselines and LLM4EA is the LLM-annotated pseudo-labels (denoted as $\mathcal{L}_a$ ). The training labels we referred to in the previous response is actually $\mathcal{L}_a$, rather than the ground truth labels. This clarification of experimental setting also answers **Q4**.
> > >
> > > - **Label ratio is relevant to the annotation budget.** The input to both the baselines and LLM4EA is denoted as $\mathcal{L}_a$, which is annotated with a budget of $\mathcal{B}=0.1|\mathcal{E}|$. In practice, the size of the pseudo-labels $|\mathcal{L}_a|$ is less than or equal to $0.1|\mathcal{E}|$ because some queries result in negative outputs (i.e., no matched target is found within the top-k), and thus no label is generated. The label size $|\mathcal{L}_a|$ scales linearly with the annotation budget $\mathcal{B}$. More pseudo-labels can be obtained by increasing this budget. We intentionally did not use a larger default budget to demonstrate how LLM4EA and the baselines perform under challenging cost-constrained settings. If more labels are required, the budget can simply be increased to improve performance, as shown in Figure 2 of our draft.
> > > - **Fair comparison by same input.** As discussed above, both baselines and the LLM4EA use the same input (same annotation budget to the LLMs to get $\mathcal{L}_a$, and no ground truth labels).

---

> > > > ### Author Response · Authors · 2024-08-13
> > > > **Response to reviewer SZjY (2)**
> > > >
> > > > Following the above answers, below are answers for the rest questions:
> > > >
> > > > **2.2 LLM is only applied in generating $\mathcal{L}_a$ and does not affect the bootstrapping process of bootEA**
> > > >
> > > > **Q5.** Methods like BootEA have their own iterative process to refine their pseudo-labels. This is also why I think the proposed method lacks novelty. The authors just replaced their decision function with an LLM.
> > > >
> > > > **Answers:**
> > > >
> > > > To clarify, we first denote the bootstrapped pseudo-labels in bootEA as $\mathcal{L}_b$, to distinct from LLM-annotated labels $\mathcal{L}_a$.
> > > >
> > > > - We respectfully point out that, when we use $\mathcal{L}_a$ as the initial training labels for bootEA, we don't change the bootstrapping process of bootEA. Therefore, when running baselines, **the generation of refinement of $\mathcal{L}_b$ is the default process and does not involve the LLM**.
> > > > - As we discussed in lines 265-268 in the draft, the pseudo-label generation and refinement of bootEA relies on the embeddings **trained on a set of labels**, as a result, **the refinement process can only be applied to $\mathcal{L}_b$, and cannot be applied to $\mathcal{L}_a$** as there is no trained embeddings at the first iteration.
> > > > - In contrast, although LLM4EA works iteratively, this process is not bootstrapping like BootEA. Its iterative process is designed to dynamically adjust the source entity selection policy. If we remove the active selection strategy, $\mathcal{L}_a$ is generated at once for LLM4EA, as for baselines. **In this case, we can still refine the label accuracy of $\mathcal{L}_a$ while bootEA cannot**, as our probabilistic-reasoning-based label refiner is **unsupervised** that only exploit the structural compatibilities among $\mathcal{L}_a$ to refine labels, without relying on ground truth labels or embeddings trained on them.
> > > >
> > > > **2.3 Annotation when without name features by exploiting textual attributes**
> > > >
> > > > **Q6.** How can "LLMs generate effective labels without name information, which justifies the 2nd claim"?
> > > >
> > > > **Q7.** If the string or word does not exist, how do you filter out the less likely counterparts?
> > > >
> > > > **Q8.** If the name feature is unavailable, how can you represent these triplets?
> > > >
> > > > **Answers:**
> > > >
> > > > When name information is not available, we leverage the associated attributes to represent each entity, these attributes are used for counterpart filtering and target entity identification. Specifically, we
> > > >
> > > > - First employ a python regular expression to remove the meaningless attributes as most of the attributes are meaningless and do not contain semantic information;
> > > > - Then we concatenate these attributes and employ a bert model to get the word embeddings, based on these embeddings we can recall the top-20 similar entities;
> > > > - Finally we adjust the prompt, and describe each entity with its associated attributes,  and apply LLM to identify the target for each source entity from its top-20 candidates.
> > > >
> > > > For more details of how to leverage attributes for filtering and target entity identification, we kindly refer 1st section of the response to reviewer n1nd in 10 Aug 2024, 18:33.
> > > >
> > > > Thank you again for the time and efforts you invested in participating in the discussion. Your comments are invaluable for helping us improve the quality of the draft and enable us to clarify the confusions.

---

> > > > > ### Comment · Reviewer_SZjY · 2024-08-13
> > > > >
> > > > > Hi, thank you very much for your detailed explanations to my questions. The authors may highlight too much on the so-called label-free setting. In fact, there are many existing methods that also have the unsupervised experiments [1, 2], in which they also leverage the similarity scores of text and vision information to produce the initial training labels. Then, they use the bootstrapping algorithm to update the training labels, which is analogous to this paper from my point of view.
> > > > >
> > > > > I still tend to rejection due to the novelty, but I understand it's difficult to conduct new experiments at this stage. Thus, I would like to raise my score from 3 to 4.
> > > > >
> > > > > [1] Multi-modal Contrastive Representation Learning for Entity Alignment. COLING, 2022.
> > > > >
> > > > > [2] MEAformer: Multi-modal Entity Alignment Transformer for Meta Modality Hybrid. MM, 2023.

---

> > > > > > ### Author Response · Authors · 2024-08-14
> > > > > > **Response to reviewer SZjY**
> > > > > >
> > > > > > Dear Reviewer SZjY,
> > > > > >
> > > > > > Thank you for acknowledging our previous explanations and for raising the rating. We are genuinely grateful for your comments, which have helped us address these concerns.
> > > > > >
> > > > > > We would like to take this opportunity to answer the remaining concern (e.g. novelty), and clarify the distinctions between our work and the methods you mentioned that also enable unsupervised EA:
> > > > > >
> > > > > > 1.  **Scalable pseudo-label generation.** While the multi-modal EA models you referenced generate pseudo-labels using feature similarities (e.g., names, images) and enhance them through bootstrapping, LLM4EA focuses on text features. In low-resource scenarios where only text attributes are available, relying solely on text embedding matching can lead to many false negatives. For example, in OpenEA D-W-15K V1 (without names), we found that using BERT-generated word embeddings results in low precision, as shown below. Precision drops further with larger KGs like D-W-100K V1 due to increased text similarity in larger datasets.
> > > > > >
> > > > > >     | Emb-match on OpenEA V1 (w/o name) | Hit@1 | Hit@20 | MRR   |
> > > > > >     | --------------------------------- | ----- | ------ | ----- |
> > > > > >     | D-W-15K                           | 0.125 | 0.237  | 0.158 |
> > > > > >     | D-W-100K                          | 0.047 | 0.107  | 0.068 |
> > > > > >
> > > > > >     In contrast, LLM4EA employs a recall-then-annotate strategy, leveraging the semantic understanding and reasoning capabilities of LLMs to annotate from the top-k candidates identified by embedding matching. This approach achieves significantly higher precision: **0.86** on D-W-15K and **0.82** on D-W-100K.
> > > > > >
> > > > > > 2.  **Technically advanced label refinement.** While our pseudo-labels are more precise than those from feature matching, they still require refinement before model training begins. Our unsupervised label refiner effectively handles this and is applicable to all kinds KGs. The methods you mentioned could benefit from incorporating our technique. Notably, while recent models like EASY [1] also offer unsupervised label refinement, their design is less effective because they 1) only consider one-hop structures, whereas LLM4EA's label refiner leverages higher-order structures through logical deduction, and 2) overlook relational properties within KGs, which LLM4EA addresses..
> > > > > >
> > > > > > 3.  **Active selection strategy.** LLM4EA also integrates an active selection strategy to tackle the unique challenges of LLM-based pseudo-label generation, particularly when annotation is costly and budgets are limited. This module optimizes utility of the annotation budget.
> > > > > >
> > > > > > We hope this explanation clarifies the unique contributions of LLM4EA and why it stands out in unsupervised EA settings.
> > > > > >
> > > > > > We also appreciate the listed methods for enhancing the comprehensiveness of the related work discussion, and we will incorporate these references in our draft revision.
> > > > > >
> > > > > >
> > > > > > [1] Make it easy: An effective end-to-end entity alignment framework. SIGIR, 2021

---

### Official Review · Reviewer_n1nd · 2024-07-07

**Soundness:** 2
**Presentation:** 2
**Contribution:** 1
**Rating:** 7
**Confidence:** 5

**Summary:**

This paper proposes a new setting that uses LLM to generate entity alignment training pairs, then use the generated pairs to train a matching model for entity alignment between two KGs. The proposed method is evaluated on the OpenEA dataset and achieves better performance than baseline methods on the setting.

**Strengths:**

A new setting for entity alignment.

The proposed method achieves better performance than baseline methods on the OpenEA dataset in the setting.

The paper is well-written and easy to follow.

**Weaknesses:**

1. The setting is impractical.
Is LLM really necessary for generating training pairs of entity alignment? if you already know the names of entities in two KGs, you can just use string matching / semantic similarity to generate training pairs. It produces much more accurate training pairs than LLM with "a fixed budget". I don't see the point of using LLM for generating training pairs of entity alignment. For the mono-lingual datasets, simply string matching can already achieve 100% accuracy, and for the cross-lingual datasets, lots of papers using semantic similarity have proven its effectiveness (https://arxiv.org/abs/2210.10436, https://aclanthology.org/2022.acl-long.405/, https://www.ijcai.org/proceedings/2020/439, https://dl.acm.org/doi/abs/10.1145/3404835.3462870, https://arxiv.org/abs/2203.01044, https://aclanthology.org/2021.emnlp-main.226/).


2. Line 237 "We have chosen "V2" because it more closely resembles existing KGs". This statement is not true. In V2's generation process, they first randomly delete entities with low degrees, which results in a denser graph. This is not the case for existing real-world KGs, which are usually sparse. In their original paper, they said that the V2 benchmark is "more similar to existing datasets" like DBP15k, these datasets are not real-world KGs, they are synthetic datasets. Thus, the statement is misleading and should be corrected.

3. Following the above point, the proposed method is evaluated on the OpenEA V2 dataset, which is twice denser than real-world KGs. The performance on the OpenEA V2 dataset may not be representative of the performance on real-world KGs. This raises the question of the generalization of the proposed method to real-world KGs.


4. Baselines are old, see point 1 for more details.

5. The setting can actually be seen as a special case of the setting in https://dl.acm.org/doi/10.1145/3394486.3403268. The difference is the noise's source, in this paper, the noise comes from LLM, in the other paper, the noise is manually added. Mitigating noise from the training set/textual label is not new, they can be found in point 1's references.

6. As far as I know, the DBpedia, Wikidata, and YAGO datasets are all from the same domain, which gives them identity entity names. Take D_Y_15K_V2 for example, if you calculate the edit distance between the entity names in these datasets, you will find that the same entity in different datasets has exactly 0 edit distance. How exactly is Table 7 obtained? Table 7 shows the performance of semantic similarity/string matching is a lot worse than the proposed method and the F1 can be as low as 2% in some cases. This is not true, the F1 of semantic similarity/string matching should be 100% in the monolingual case.  The results in Table 7 are not reliable and should be corrected. I suggest the authors to check the correctness of the dataset and the code implementation.

Overall, this paper contains interesting but impractical new settings, false statements, and unreliable results. I recommend rejection.

**Questions:**

See above

**Limitations:**

A limitations section is provided in the paper

---

> ### Author Rebuttal · Authors · 2024-08-07
>
> We respectfully thank you for the helpful feedback that helps improve the draft. We answer the concerns and address weaknesses (w1-w6) as below:
>
> # [w1, w4] Reasonability of using LLMs in EA and comparison with recent methods
>
> We sincerely thank for pointing out this subtle point, and we answer this question from following two aspects:
>
> 1.   **Potential in more challenging scenario.** Although we perform evaluation in a simple scenatio (entity names are available) in the experiments, the potential of using LLMs in EA tasks is more than that. Specifically, **language models can annotate challenging datasets where no entities names are available**, by exploiting the semantic information within textual attributes, where **existing name-based methods (e.g. LightEA, SEU, BERT-INT) cannot handle**. To empirically evaluate this, we entity alignment on D_W_15K_V1 with LLM4EA, where the target entities are IDs to avoid name bias. Results are as below,
>
>      | Alignment performance | Annotation precision    |
>      | -- | - |
>      | 0.472 (hit@1) | 301 (true positive) |
>      | 0.654 (hit@10)   | 49 (false postive)  |
>      | 0.536 (MRR)  | 1150 (abandoned)  |
>
>      As shown in the results, LLMs can generate effective pseudo-labels without the need for identical names and get promising results.
>
> 2.   **Scalability to larger KGs.** The target of entity alignment is building a larger unified KG. Although existing finetuning-based models (BERT-INT, SDEA, CEA) can also leverage semantic information, our incontext-learning based framework 1) does not require pre-aligned labels to finetune; and 2) is more scalable as it does not require local hardware configuration for LLMs.
>
> we appreciate the listed methods for boosting the comprehensiveness of the discussion of related work, and we will add these results in the draft revision.
>
> # [w2 & w3] Results on OpenEA V1
>
> Thank you for detailed reviews, we will correct the false statement about dataset version in the revision. To validate the performance on real-world KGs, we attach the experimental results on OpenEA V1 dataset as below. Due to character limit, we only show the results of strong&new baselines.
>
> |    | D_W  | (w/o ent  | name)  |  | D_Y  |   |   | EN_DE |  |  | EN_FR  |  |
> | -- | - | -- | - | - | - | - | - | -- | -- | --- | - | - |
> |     | hit@1     | hit@10    | mrr       | hit@1     | hit@10    | mrr    | hit@1     | hit@10    | mrr       | hit@1     | hit@10    | mrr       |
> | BootEA     | 0.302     | 0.584 | 0.413     | 0.664   | 0.811  | 0.718     | 0.655     | 0.844     | 0.720     | 0.429     | 0.685     | 0.515     |
> | Dual-Amn    | 0.386     | 0.639 | 0.479     | 0.748   | 0.858  | 0.790     | 0.728     | 0.910     | 0.793     | 0.537     | 0.800     | 0.628     |
> | RDGCN    | 0.337     | 0.436  | 0.372     | 0.817  | 0.923  | 0.858     | 0.646     | 0.775     | 0.692     | 0.597     | 0.742     | 0.649     |
> | LLM4EA    | **0.472** | **0.654** | **0.536** | 0.755  | 0.859     | 0.795     | **0.736** | **0.918** | **0.800** | 0.581     | **0.819** | 0.662     |
> | LLM4EA (RDGCN) | 0.421  | 0.532  | 0.476  | **0.842** | **0.931** | **0.883** | 0.672     | 0.787     | 0.736     | **0.642** | 0.761     | **0.683** |
>
>
>
> As shown by the results, LLM4EA consistently outperform strong baselines. It is noteworthy that LLM4EA is a general framework and can employ any base EA model such as RDGCN (last row) to perform robust label-free entity alignment with LLMs.  We argue that **the conclusions and claims holds on this sparse dataset**. And we will include the updated results in the draft revision.
>
> # [w5] Comparison with existing work that addresses noise labels (REA)
>
> The paper referenced in weakness 5 (REA) also addresses noise in the training set. However, our work is significantly distinct from theirs in three key aspects: **1) Technical novelty**. The noise detection model in REA is trained as a binary classifier using an adversarial training paradigm, In contrast, our label refiner employs probabilistic reasoning for robust label refinement. **2) Flexibility.** LLM4EA can leverage any off-the-shelf EA models as base models without altering the model architecture or training objectives. **3) Superior performance**. We empirically tested the performance of REA on annotated pseudo-labels generated by LLMs and present the results below and show the superiority of LLM4EA.
>
> |        |          D_W       | (w/o ent  | name)            |           | D_Y       |           |           | EN_DE     |           |           | EN_FR     |           |
> | ------ | ---- | ----- | --------- | --------- | ------ | ------ | --------- | --------- | --------- | --------- | --------- | --------- |
> |        | hit@1     | hit@10             | mrr       | hit@1     | hit@10    | mrr       | hit@1     | hit@10    | mrr       | hit@1     | hit@10    | mrr       |
> | REA    | 0.213     | 0.467              | 0.298     | 0.447     | 0.798     | 0.563     | 0.363     | 0.742     | 0.475     | 0.226     | 0.536     | 0.352     |
> | LLM4EA | **0.472** | **0.654**          | **0.536** | **0.755** | **0.859** | **0.795** | **0.736** | **0.918** | **0.800** | **0.581** | **0.819** | **0.662** |
>
>
>
> # [w6] Low recall/f1 on semantic matching
>
> Thank you for the detailed review and constructive suggestions. We identified that the low recall was due to the word embedding model not generating embeddings for out-of-vocabulary names. We addressed this by replacing it with a pretrained language model (bert-base-uncased) and implementing a filtering process that only considers 1-1 matching. The updated results are as follows:
>
> |       | Precision | Recall | F1    |
> | ----- | --------- | ------ | ----- |
> | D-W   | 0.918     | 0.624  | 0.743 |
> | D-Y   | 1.0       | 1.0    | 1.0   |
> | EN-DE | 0.897     | 0.695  | 0.783 |
> | EN-FR | 0.889     | 0.736  | 0.805 |
>
> We hope our response can satisfactorily address your concerns and we would appreciate if you could consider raising the score.

---

> ### Comment · Reviewer_n1nd · 2024-08-07
>
> Thank you for your response and your effort to address my concerns. I appreciate your time and effort. I have raised my score to 4. This is because some parts of my concerns aren't fully addressed. Regarding the motivation, I hope you can address these in further responses:
>
> 1.
> The ID experiment is interesting. Please provide more detailed settings for this experiment. For example, how do you ensure that the IDs are not memorized by the LLMs? Does the accuracy come from the IDs already being memorized by the LLMs as part of the knowledge base? If so, how can we ensure that the LLMs can identify IDs that are not memorized? Wouldn't this memorization lead to more bias than name-bias issues?
> If we want to use such systems in applications where the resources are so limited that even names are not available, how can we ensure that the LLMs can still identify the IDs?
> Could you provide a more detailed experiment on this? For example, reassigning the IDs to the entities in the knowledge base and seeing if the LLMs can still identify the IDs. I assume you may need to design specific prompts, for example, showing the neighborhood IDs, to explore the reasoning ability beyond memorization
> 1.
> Scalability to larger KGs. Do you have experimental results on larger KGs such as DBP1M? Using an API to query may eliminate the need for local machines, but the API is still costly and there are rate limits. Do you have a cost analysis of using the API compared to deploying a BERT model locally? I would assume that the BERT solution is far more cost-effective and scalable, while still maintaining decent accuracy.
>
> I would advise the authors to compare the recent SOTAs such as LightEA (https://arxiv.org/pdf/2210.10436), both in terms of accuracy and scalability. Some other works like EASY (https://dl.acm.org/doi/abs/10.1145/3404835.3462870) also have some kind of error correction mechanism. It would be interesting to discuss how your work is different from these works.

---

> > ### Author Response · Authors · 2024-08-10
> > **Response to answer remaining concerns**
> >
> > Thank you for your prompt reply and support, we anwer your remaining conerns one by one below.
> >
> > # 1. Concerns about ID leakage
> >
> > Below we attach the detailed experimental setting and results to answer the concern.
> >
> > **1.1 Experimental setting**
> > - **ID random reassignment for eliminating name bias**. When masking entity names with IDs, we randomly reasign new IDs (rather than the original IDs extracted from wikidata dump file) to the entities of D-W-15K-V1.
> > - **Attribute-based prompting**. Since the entity names are not available, we describe each entity by its associated attributes in the prompt. Noteworthy that most attributes don't contain semantic information, we use a python regular expression to filter out meaningless attributes.
> >
> > **1.2 Experimental results**
> >
> > Entities are annotated in two steps.
> >
> > -   Counterpart selection
> >
> > Using the attributes, we first employ a BERT to generate embedding for each entity, then get the top-20 counterparts by semantic matching. The recall (hit@k) of this process is:
> >
> > |Hit@1|Hit@20|MRR|
> > |-|-|-|
> > |0.125|0.237|0.158|
> >
> > -   Target identification
> >
> > Given a source entity and its top counterparts, an LLM determines if an aligned target entity exists in these candidates. If the LLM finds no match, the query is discarded. Otherwise, it predicts a positive label (true or false positive). The results below are based on 1500 queries.
> >
> > | Abandoned | True positive | False positive |
> > | - | - | - |
> > | 1150 | 301 | 49 |
> >
> > # 2. Comparison with BERT model in terms of scalability
> >
> >  Below we compare LLM4EA and Bert models in terms of cost scalability and performance scalability.
> >
> > **2.1 Cost scalability**
> >
> > -   **LLM4EA's costs scale linearly with KG size.** The costs of LLM4EA mainly come from its query to LLMs. This cost scales linearly with the KG size. In our experiments, the average token cost for each entity is around 1100. Thus, each experiment on 15k dataset costs 1100x(0.1x15000)x0.5/10^6 =0.825 dollars, and for each experiment on 100k, it costs 5.5 dollars, under the pricing scheme of gpt-3.5-turbo-0125.
> > -   **Querying LLMs can be accelerated by parallel or batch query**.  Acceleration can be achieved by parallel query,  or by querying OpenAI's batch API. Noteworthy that the LLMs response speed is growing rapidly and we can benefit from these advances.
> >
> > **2.2 Performance scalability**
> >
> > Given the hardware demands and need for pre-aligned pairs in finetuning, an alternative is to use BERT for embedding matching without finetuning. In previous emb-match experiments, BERT performs well on small datasets with names, but **the following experiments show its precision decreases as KG size increases**.
> >
> > |DBP1M |Precision|Recall|F1|
> > |-|-|-|-|
> > |$DBP_{EN-DE}$|0.492|0.679|0.571|
> > |$DBP_{EN-FR}$|0.490|0.654|0.560|
> >
> > |OpenEA V1 (w/o name)|Hit@1|Hit@20|MRR|
> > |-|-|-|-|
> > |D-W-15K|0.125|0.237|0.158|
> > |D-W-100K|0.047|0.107|0.068|
> >
> > - **Bert is less accurate on large KGs**. The DBP1M dataset contains name information, while BERT model shows decreased precision. We investigate and identify that **this precision decline is mainly due to the increased number of similar names as the KG size grows**. And as expected, when name information is not available in D-W dataset (second table above), the precision also decreases.
> >
> > - **LLMs generate more precise labels**. Bert model can help recall the possible counterparts but fail to generate effective labels, employing a LLM can annotate more precise labels from these recalled counterparts, as discussed and shown by experiments in **1.2**.
> >
> > # 3. Comparison with related work
> >
> > **3.1 Comparison with LightEA**.
> >
> > |results on $DBP_{EN-DE}$|hit@1|Hit@10|mrr|
> > |-|-|-|-|
> > |LightEA  | 0.055 |0.089 |0.067|
> > |LLM4EA(LightEA)|0.099|0.152 |0.117|
> >
> > |results on $DBP_{EN-FR}$|hit@1|Hit@10|mrr|
> > |-|-|-|-|
> > |LightEA|0.034|0.066|0.045|
> > |LLM4EA(LightEA)|0.044|0.086|0.059|
> >
> > - **LightEA is efficient.** LightEA performs EA with notable efficiency, scalable to DBP1M.
> > - **LightEA can be enhanced by LLM4EA and in return improve scalability of LLM4EA.** We respectfully point out that, LLM4EA is a general framework and can incorporate any EA model as its base model to perform effective learning. In return, LLM4EA stands to benefit from advancements in efficient base EA models.
> >
> > **Comparison with EASY**.
> >
> > - **EASY only consider one-hop structure while LLM4EA leverage higher-order structures.** EASY generates confident pseudo-labels by selecting entity pairs with high (one-hot) neighborhood similarities. In contrast, LLM4EA incorporates iterative reasoning that **implicitly performs multi-hop reasoning** through logical deduction. Realworld KGs can be sparse and contain many long-tail entities that have no aligned neighbors.
> > - **EASY neglects relational properties while LLM4EA does**. Statistical properties of relations, such as functionalities, are crucial in quantifying the contributions of neighbors. EASY simply relying on neighbor counts and overlooks these properties.

---

> > > ### Comment · Reviewer_n1nd · 2024-08-11
> > >
> > > Thank you for your response. I have carefully read your response, and have the following questions:
> > >
> > > Is the label for DBP1M also from LLM? The performance of LightEA is pretty different compared with reported in the original paper. I assume it is the same 'noisy label' from LLM. Could you please confirm this and provide LLM label statistics as in 1.1?
> > >
> > > In experiment 3.1 LLM4EA has a hit@1 of 0.04~0.1, and is much lower than the simple BERT solution. Could this mean a 10% LLM annoation is not enough for larger datasets, and the annoation percentage may also be scaled with the dataset size? Does the cost analysis in 2.1 consider this?
> > >
> > > Larger dataset could result in more similar entity names, as shown in the response. This is intuitive. How LLM4EA handles this issue? If Levenshtein distance is used for filtering, won't it be worse for larger datasets?
> > >  Could you provide the recall of the filtering step?
> > >  Have the authors tried more complex blocking methods in entity resolution to improve the performance?
> > >
> > > Could you detail the prompt template used for both the name experiment and the ID experiment?  The prompt template is crucial for reproducibility, and it is important to provide it in the paper, at least in the appendix.

---

> > > > ### Author Response · Authors · 2024-08-11
> > > > **Responses to Reviewer  n1nd**
> > > >
> > > > Thank you for your prompt reply and detailed questions. Below are our responses:
> > > >
> > > > **1. Labels for DBP1M and statistics**
> > > >
> > > > The labels for both LightEA and LLM4EA(LightEA) were annotated by LLMs, with a precision of approximately 0.861 for $DBP1M_{EN-DE}$ and 0.830 for $DBP1M_{EN-FR}$. During our experiments with LLM4EA(LightEA), these precision rates were improved to 0.94 and 0.91, respectively.
> > > >
> > > > **2. Compromised performance comes from reduced entity dimension**.
> > > >
> > > > During our experiments, we found that LightEA could only run DBP1M on a **GPU** with an entity dimension of `ent_dim=64`. Running with higher dimensions, such as `ent_dim=256`, led to out-of-memory errors, despite claims in the original paper that `ent_dim=256` was feasible. We successfully ran $DBP1M_{EN-DE}$ with `ent_dim=256` on a **CPU**, with results as follows:
> > > >
> > > > | Model           | hit@1 | hit@10 | Mrr   |
> > > > | --------------- | ----- | ------ | ----- |
> > > > | LightEA         | 0.245 | 0.336  | 0.293 |
> > > > | LLM4EA(LightEA) | 0.272 | 0.394  | 0.314 |
> > > >
> > > > Running these experiments on a CPU took approximately two days, therefore we did not further run it on $DBP1M_{EN-FR}$ due to limited time. Based on the results above, we can see
> > > >
> > > > -   **The annotation percentage still scale to large datasets**. The results with dimension 256 are already quite approximate to the results in the LightEA paper although we only annotate 10% of the nodes.
> > > >
> > > > -   **The compromised performance in previous results was mainly due to the reduced entity dimension**. The choice of entity dimension does not affect our claim that "LLM4EA can enhance LightEA," as our experimental settings ensure a fair comparison. We will clarify this subtle point in the revised paper and include the results on $DBP1M_{EN-FR}$ with `ent_dim=256`.
> > > >
> > > >
> > > > **3. LLM4EA handle similar names by recall-then-annotate strategy.**
> > > >
> > > >  As detailed in **1.2** in our previous response, LLM4EA first performs counterpart filtering to ensure recall rate, then identifies with the semantic understanding and reasoning ability of LLM4EA.
> > > >
> > > > For the counterpart filtering process, the recall rate (in top-20) of $DBP1M_{EN-FR}$ and $DBP1M_{EN-DE}$ is 0.791 and 0.834. This recall rates show that the current string distance-based filtering method works well for our task, therefore we didn't incorporate more intricate filtering methods. When more complicated scenarios are present, we can also benefit from more sophisticated filter such as blocking, scaling our framework to such scenarios. We appreciate your comments and suggestions that extend the discussion of possible challenges in practice.
> > > >
> > > > **4. We will provide the detailed prompt in the paper**
> > > >
> > > > Thank you for this suggestion. We will include the detailed prompts in the appendix of the revised paper.
> > > >
> > > > We sincerely thank you again for the time and effort you invested in reviewing our response. Your comments are invaluable for improving our work and enable us to clarify any confusions.

---

> > > > > ### Comment · Reviewer_n1nd · 2024-08-11
> > > > >
> > > > > Thank you for your response. Based on the new experiments and discussions provided by the authors, I have adjusted the score. I would appreciate it if these updates could be incorporated into the next version of the paper.

---

> > > > > > ### Author Response · Authors · 2024-08-12
> > > > > > **Response to Reviewer n1nd**
> > > > > >
> > > > > > Dear Reviewer n1nd,
> > > > > >
> > > > > > We are delighted that our work has been acknowledged by you following these discussions. We sincerely thank you once again for the time and effort you invested in reviewing our work and engaging in the discussion. Your professional comments are invaluable for improving our work, and your active participation has enabled us to clarify any confusions. We sincerely appreciate your contribution.

---

### Official Review · Reviewer_GwVz · 2024-07-12

**Soundness:** 3
**Presentation:** 3
**Contribution:** 3
**Rating:** 7
**Confidence:** 2

**Summary:**

This paper tackles the challenge of entity alignment (EA) in merging knowledge graphs (KGs) by leveraging Large Language Models (LLMs) to automate annotations, addressing the costly and impractical reliance on human-generated labels despite difficulties like large annotation space and noisy labels. They propose LLM4EA, a unified framework designed to harness LLMs effectively for EA tasks. Key contribution include a novel active learning policy that reduces the annotation space by prioritizing the most valuable entities based on the overall inter-KG and intra-KG structure, and an unsupervised label refiner that enhances label accuracy through probabilistic reasoning. The framework iteratively optimizes its policy using feedback from a base EA model. Experimental results on four benchmark datasets highlight the framework's effectiveness, robustness, and efficiency, showcasing its potential as a significant advancement in the field of entity alignment.

**Strengths:**

- This paper proposes an iterative annotation framework leveraging the capability of LLMs' zero-shot in-context learning (ICL). The proposed framework, using an iterative refinement strategy, can effectively reduce costs by utilizing cheaper LLMs such as GPT-3.5.
- The experimental design of the paper is reasonable and sufficiently demonstrates the effectiveness of the proposed framework.
- This paper is well-written and easy to follow.

**Weaknesses:**

- The authors do not consider any open-source LLMs such as Llama2/3. For example, Llama3 8b/70b may have better performance in EA tasks compared to GPT-3.5.
- The authors did not report the actual API costs for the experiments. For the baselines, what budgets were used to annotate the dataset to train these models? Are they the same as those used in your iterative framework?
- Figure 1: Replacing the human icon with a robot for the LLM annotator may be better.

**Questions:**

Please see the weakness.

**Limitations:**

This paper has one limitation section.

---

> ### Author Rebuttal · Authors · 2024-08-07
>
> Thank you for your helpful feedback that helps us improve the draft. We answer concerns and address the three weaknesses (w1, w2, w3) in the following.
>
> **[w1] Ablation of LLM model**
>
> The choice of LLM model can affect annotation accuracy and cost. We recognized this factor and employed both GPT-3.5 and GPT-4 as annotators, comparing the differences (Table 1 & Figure 2 in the draft) between less powerful and more powerful LLMs. Thank you for the suggestion; we will include experimental results of additional LLMs, especially open-source ones, to complement the empirical analysis.
>
>
> **[w2] Cost measurement**
>
> As we have stated in the experimental setting (line244-245), baselines and our framework use the same budget (line239), which is $0.1\times|\mathcal{E}|$. To ensure a fair comparison, we use the same prompt template for all experiments. This also ensures that the actual API cost for baselines and our framework is statistically the same (although a slight variance brought by the different lengths of entity names). Based on our experimental observation, the average API cost for annotating each entity is around 1100 tokens, thus it takes around 1100x(0.1x15000)x0.5/10^6 =0.825 dollars to run each alignment task using gpt-3.5-turbo-0125.
>
>
> **[w3] Improvement of presentation.**
>
> Thank you for the detailed review, we will replace the icon in figure1 in the draft revision.

---

> > ### Comment · Reviewer_GwVz · 2024-08-09
> > **To Authors**
> >
> > Thanks for your responses. Your answers address most of my concerns.
> > I think my rating is reasonable and fair. So I maintain my scores.

---

### Official Review · Reviewer_GZQy · 2024-07-27

**Soundness:** 3
**Presentation:** 2
**Contribution:** 3
**Rating:** 6
**Confidence:** 3

**Summary:**

The paper proposes an active learning/weak supervision based approach for knowledge base alignment (aligning entities across KGs). The paper explores an LLM labeler for generating entity alignment labels but explores active selection of source nodes labeled by the LLM, a label refiner which refines the LLM annotations based on the structural consistency of the KGs (nodes connected to aligned nodes should be aligned), and finally trains a entity alignment model on the gathered data. The paper compares to a range of baselines and presents several ablations to indicate the efficacy of the proposed approach.

**Strengths:**

- The paper proposes an interesting approach which seems novel and applied to a meaningful problem.
- The paper's experiments are thorough.

**Weaknesses:**

- The papers writing is hard to follow and could benefit from reduction in notation, addition of intuition for modeling choices, and clearer distinguishing of its own contributions wrt prior work.

**Questions:**

NA

---

> ### Author Rebuttal · Authors · 2024-08-07
>
> Thank you for your time and effort in reviewing our work. We appreciate your insights and will address these points. Below is a summary of the content of our work.
>
> - **Motivations.** Our work is motivated by the fact that existing methods heavily rely on accurate seed alignment pairs for training. Annotating such pairs requires substantial cross-domain knowledge, making it very costly. Large Language Models (LLMs) offer new opportunities in automating this task by processing semantic information and generating pseudo-labels.
> - **Challenges.** Employing LLMs for this task is nontrivial because the search space is vast, and LLMs can generate false labels, which can harm the final performance. There is no existing framework to handle these challenges effectively.
> - **Contributions**. As itemized at the end of the introduction section (lines 69-84), our contributions include 1) A novel LLM-based in-context learning framework for label-free entity alignment; 2) An unsupervised label refiner to enable effective training on noisy pseudo-labels; and 3) An active sampling module to maximize the utility of the annotation budget.
>
> We sincerely thank you for your recognition and helpful feedback. We will further polish the paper presentation based on your comments in the revision.

---

### Decision · Program_Chairs · 2024-09-25

**Decision:**

Accept (poster)

**Comment:**

The paper proposes a way to use LLM for Entity Alignment tasks (LLM2EA). It introduced an active learning policy to help reduce the annotation space, which is a unique challenge to vanilla LLMs in EA tasks. It also adds an unsupervised label refiner to efficiently collect pseudo-labels. Experiments showed clear improvement over other methods, including the additional ones suggested during rebuttal.

A main concern about the paper is around novelty (Reviewer SZjY), that it's similar to the core idea of previous methods like BootEA. The authors provided new evaluation data to show that LLM2EA has significantly better results than BootEA. AC acknowledges the concern of SZjY, however given the delta in experiment metrics and the differences explained by the authors, the delta seems non-trivial.